# General lightweight framework for vision foundation model supporting multi-task and multi-center medical image analysis

Senliang Lu [1,2,3,11], Yehang Chen [1,11], Yuan Chen[4,11], Peijun Li[5], Junqi Sun[6], Changye Zheng[7], Yujian Zou[7], Bo Liang[8], Mingwei Li[9], Qinggeng Jin[10], Enming Cui [2,5], Wansheng Long [2,5] ✉ & Bao Feng [1,2] ✉

The foundation model, trained on extensive and diverse datasets, has shown strong performance across numerous downstream tasks. Nevertheless, its application in the medical domain is significantly hindered by issues such as data volume, heterogeneity, and privacy concerns. Therefore, we propose the Vision Foundation Model General Lightweight (VFMGL) framework, which facilitates the decentralized construction of expert clinical models for various medical tasks. The VFMGL framework transfers general knowledge from large-parameter vision foundation models to construct lightweight, robust expert clinical models tailored to specific medical tasks. Through extensive experiments and analyses across a range of medical tasks and scenarios, we demonstrate that VFMGL achieves superior performance in both medical image classification and segmentation tasks, effectively managing the challenges posed by data heterogeneity. These results underscore the potential of VFMGL in advancing the efficacy and reliability of AI-driven medical diagnostics.

Identifying and segmenting medical images, such as detecting myometrial invasion in endometrial cancer[1,2], identifying breast cancer metastases in lymph node slides[3], and segmenting the prostate[4], plays a crucial role in advancing precision cancer treatment[5–7]. Artificial intelligence (AI)-based precision medicine will lead to large-scale data being used for diagnostic purposes[8–10]. Meanwhile, the heterogeneity in imaging equipment, imaging protocols, image quality, and patient demographics among different medical centers leads to challenges of data heterogeneity (DH) in medical images from different medical centers[11,12]. DH affects the stability of feature extraction, thereby influencing the generalization ability and robustness of AI models.

In recent years, in the field of natural language processing, large-scale language models have used self-supervised learning methods[8,13–15] to learn the intrinsic structural of language from large-scale text corpora, enabling large-scale language models to perform well in multiple natural language processing tasks[16–18]. Similarly, in the field of computer vision, Vision Foundation Models (VFMs) are trained based on self-supervised methods on large unlabeled datasets with different qualities and diversities of nature images, producing universal features applicable to various visual tasks[14,15], such as SAM[14] and DINOv2[15]. These universal knowledge is crucial to improve the robustness of the model.

[1]Laboratory of Intelligent Detection and Information Processing, Guilin University of Aerospace Technology, Guilin, Guangxi, China. [2]Jiangmen Key Laboratory of Artificial Intelligence in Medical Image Computation and Application, Jiangmen Central Hospital, Jiangmen, Guangdong, China. [3]School of Electronic Engineering and Automation, Guilin University of Electronic Technology, Guilin, Guangxi, China. [4]Department of Gynecology, Jiangmen Central Hospital, Jiangmen, Guangdong, China. [5]Department of Radiology, Jiangmen Central Hospital, Jiangmen, Guangdong, China. [6]Department of Radiology, Yuebei People's Hospital, Shaoguan, Guangdong, China. [7]Department of Radiology, Affiliated Dongguan Hospital, Southern Medical University, Dongguan, Guangdong, China. [8]Department of MRI, Maoming People's Hospital, Maoming, Guangdong, China. [9]Department of Gynecology, Kaiping Central Hospital, Kaiping, Guangdong, China. [10]School of Electrical Engineering, Guangxi University, Nanning, Guangxi, China. [11]These authors contributed equally: Senliang Lu, Yehang Chen, Yuan Chen. ✉e-mail: jmlws2@163.com; fengbao1986.love@163.com

In the field of medicine, effective universal knowledge can effectively mitigate the interference of individual patient differences on model performance[8,15]. However, due to significant differences between natural images and medical images, when the aforementioned VFMs are directly applied to medical image analysis, there may be interfering features, leading to suboptimal performance of these VFMs in medical image analysis[19–22]. Jun Ma et al.[19] fine-tuned MedSAM using more than 20 A100 GPUs with a total capacity of 80GB on an annotated dataset containing over 1 million medical images. As a semi-automatic segmentation method based on prompt boxes, it demonstrated excellent performance across various medical image segmentation tasks. Eugene Vorontsov et al.[23] pre-trained the Virchow model on 1.5 million whole-slide pathology images. They then built a classifier based on model embeddings and nearly 80,000 annotated medical data samples, achieving outstanding performance on specific downstream tasks. Training, fine-tuning[24], and deploying VFMs typically require extensive data support(including annotated data) and consume substantial hardware resources and time. This severely limits the cross-center research and popularization of VFMs. Therefore, the medical field urgently needs a technology to acquire universal knowledge from VFMs, enabling models to reduce training and deployment complexity while ensuring accuracy.

Inspired by the challenges and technologies mentioned above, we have developed a Vision Foundation Model General Lightweight (VFMGL) framework to adapt to various specific downstream medical tasks. In the case of multi-center DH, VFMGL (1) can adaptively acquire medical task-related general knowledge from VFM, achieving the lightweighting of VFM; (2) maintains data privacy and facilitates collaborative training of models across multiple medical institutions; (3) is suitable for various medical tasks in classification and segmentation; (4) exhibits robustness and cross-center generalization; and (5) provides explainability at both the feature and model levels.

Addressing the issue of heterogeneous multi-center medical data and the utilization of foundation models in the medical field, we propose the VFMGL framework (Fig. 1a). VFMGL offers a method called Heterogeneous-model General Knowledge Transfer (HGKT), which automatically identifies and transfers general knowledge suitable for tasks at each center from open-source VFM to build lightweight local models with robust feature extraction capability. To further improve the cross-center generalization performance of local models, a Data Deduction in Batch Level (DDBL) approach is applied to select low-heterogeneity data from each center. Combined with Knowledge Distillation (KD), it drives the redundant parameters of local models to learn common knowledge from the shared model.

Given the significant domain differences between natural and medical images, fixed matching of model layers for knowledge transfer may not adapt well to various medical tasks across multicenter heterogeneous data, potentially leading to negative transfer[25]. The HGKT technique leverages general knowledge from open-source VFM to perform medical tasks, automatically matching model layers and identifying relevant knowledge between open-source VFM and local models. This facilitates knowledge transfer between heterogeneous models, enabling lightweight local models to extract robust visual features. HGKT establishes a feature transfer pairs, selecting general knowledge through computation of transfer weights to construct robust key layers in local models (Fig. 1b).

Due to the DH across multiple centers, local models often perform well on their own center's data but struggle with data from other centers. Studies have shown that neural networks are highly capable of capturing feature patterns specific to a particular dataset to enhance model performance[26,27]. VFMGL supports strict preservation of medical data within each center, utilizing federated learning (FL) technology[28,29] to transmit local model parameters and aggregate shared model across multiple centers, ensuring the privacy and security of local data. Shared model parameters are distributed to each center through server. Local models can easily fit private data with distinctive feature expressions; however, similar samples may be scarce in other centers, leading the shared model to lack sufficient knowledge for predicting such samples and widening the predictive distribution discrepancy between shared and local models. Based on this consideration, we propose the DDBL method, which selects low-heterogeneity data from each center based on shared model knowledge. Combined with a KD approach driven by model logical layer outputs[30], DDBL enables local models to learn common knowledge possessed by multiple centers, suppressing their tendency to learn specific feature patterns while using redundant parameters to further enhance cross-center generalization capability (Fig. 1c).

## Results

### VFMGL to Identify Myometrial Invasion in MRI for Endometrial Cancer

Endometrial Cancer (EC) is a common malignant tumor in the female reproductive system and the sixth leading cause of cancer related deaths in women[31,32]. Myometrial invasion(MI) is one of the most important prognostic factors in EC[33]. Diagnosing the presence or absence of MI aids in pre-treatment stratification, including determining the treatment approach (whether fertility-sparing is feasible), defining the surgical scope such as the necessity of lymph node dissection, and predicting prognosis[2]. We collected real clinical EC data from six hospitals as the first use case, comprising a total of 1267 patients who underwent total hysterectomy due to EC. Except for center E and F, samples from each center were divided into training and testing sets in a 6:4 ratio using a random seed of 42 (Supplementary Table 6), while center E and F serve as two external testing sets. Each center could only use open-source VFM, shared model, and training set data to train local models, and data could not be transferred between centers to simulate a multi-center clinical application scenario (unless otherwise specified, we used this default setting).

We trained local models based on the VFMGL framework for each center and achieved favorable testing performance. The Area Under the Curve (AUC) values for the test sets at each center were 0.798, 0.833, 0.857, and 0.848, respectively. To assess the diagnostic performance of VFMGL, we compared it with four FL methods, namely FedAvg[28], FedProx[34], HarmoFL[12], and MetaFed[35], as well as a vision foundation model called Virchow[23]. The results indicate that VFMGL achieved average AUC improvements of 8.9%, 6.6%, 11.4%, and 5.8% across different centers. Furthermore, VFMGL achieved an overall prediction accuracy of 0.799 (349/437) in distinguishing MI from non myometrial invasion (NMI), thus enhancing early diagnosis rates and offering hope for preserving fertility and improving prognosis for patients. Fig. 2A, B respectively illustrate the Receiver Operating Characteristic (ROC) curves and Decision Curve Analysis (DCA) curves for the five methods across the four centers. As shown in Fig. 2C, experimental results indicate that VFMGL achieved the highest overall performance and net benefit across all four centers. More details of the results can be found in Supplementary Table 1.

Considering the future scenario where independent centers lacking independent diagnostic capabilities and not participating in VFMGL training need to use the VFMGL models for disease diagnosis, we further validated the predictive performance of VFMGL on an independent external validation center E (Table 1). We estimated the similarity of data features between two centers based on data deduction, with the similarity scores between each center and the external validation center as follows: Center A: 0.15; Center B: −0.05; Center C: −0.06; Center D: 0.08. Therefore, we selected the local model from Center A for predictions on this external validation set. The results showed that the AUC for Model A was 0.742 (Model B: 0.694; Model C: 0.661; Model D: 0.710), and the ROC curves for each

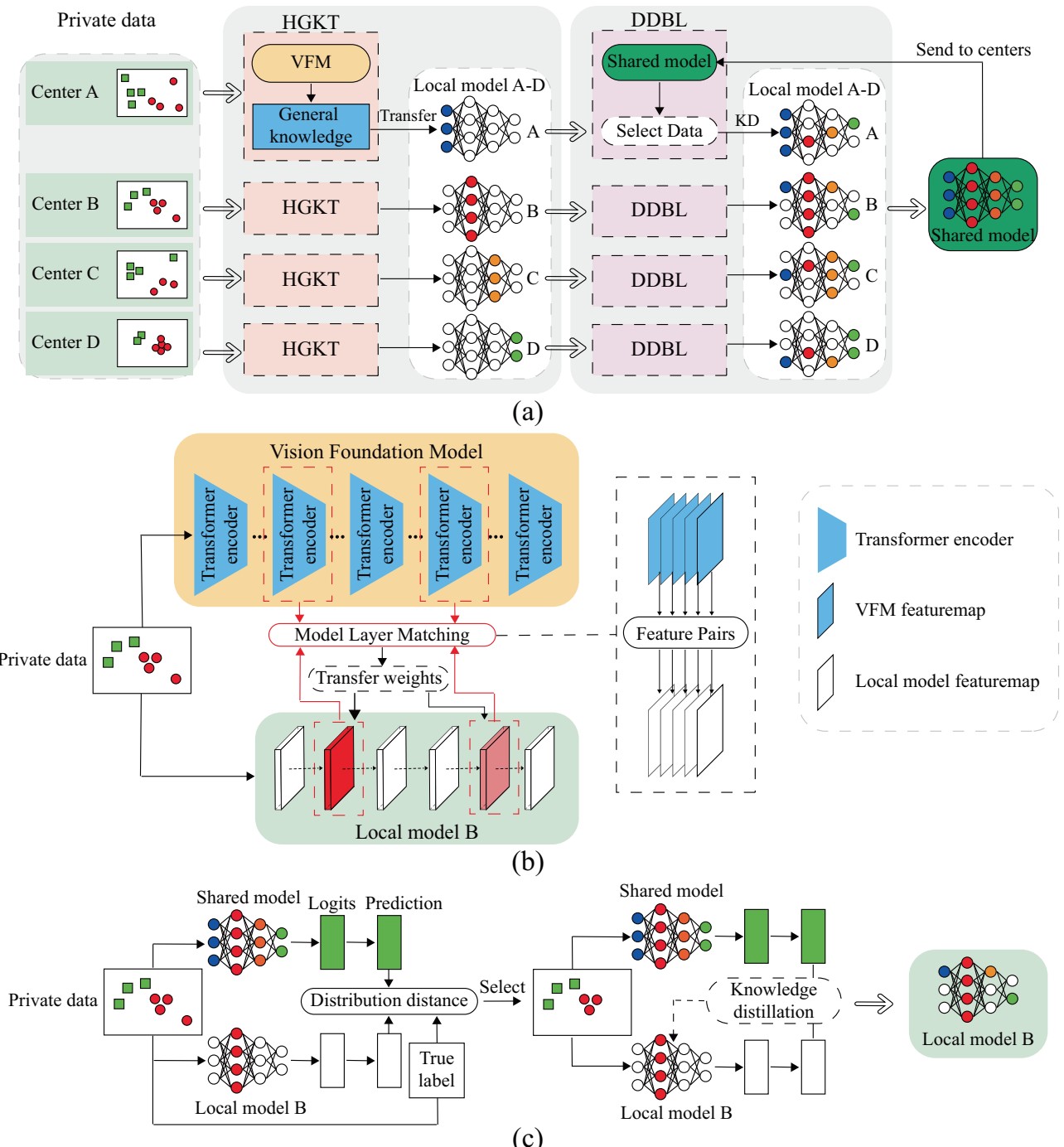

**Fig. 1 | Overview of the VFMGL framework. a** Construction of Local and shared Models. **b** Construction of robustness critical layers based on HGKT and general knowledge from VFM. **c** Continued model construction based on DDBL and common knowledge. VFMGL Vision Foundation Model General Lightweight, HGKT Heterogeneous model General Knowledge Transfer, DDBL Data Deduction in Batch-Level, VFM Vision Foundation Model.

model are illustrated in Supplementary Fig. 1. Additionally, we collected 117 cases from a new medical center, F, as an external validation set 2 (see Table 1) to further evaluate the performance of VFMGL. Based on the aforementioned approach, we calculated the data features similarity between centers A–D and center F, with values of −0.04, 0.12, −0.11, and 0.01, respectively. The results show that model B achieved an AUC of 0.720 (model A: 0.680, model C: 0.601, model D: 0.702). The ROC curve is shown in Supplementary Fig. 13. These findings indicate that VFMGL maintains relatively stable predictive performance on data from a new center that did not participate in information exchange.

## Identification of breast cancer metastasis in lymph node sections

Breast cancer is the most common cancer among women in the United States[31]. Approximately 12% of women are diagnosed with breast cancer in their lifetime[36]. Axillary lymph nodes are typically the first site of breast cancer metastasis, and identifying metastases in lymph nodes holds therapeutic significance for breast cancer patients[37]. Whole-slide images (WSIs) digitize high-resolution slide pathology, enabling AI to assist in this time-consuming and tedious pathology examination, thus improving the efficiency and accuracy of histopathological lymph node assessments. However, due to differences in slide staining and

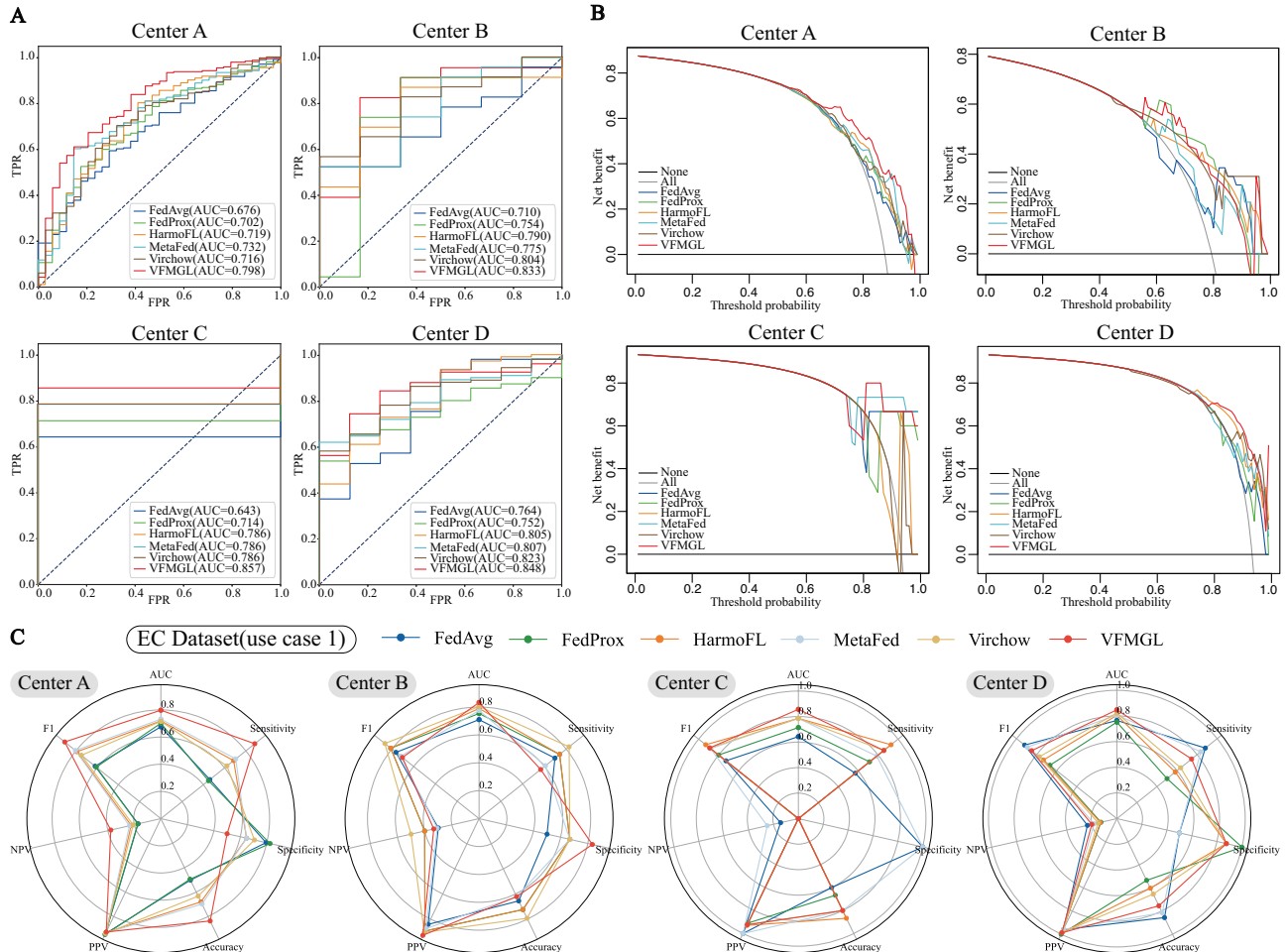

**Fig. 2 | ROC curves, DCA curves and radar charts for the four centers. A** ROC curves of five models in four centers. **B** DCA curves of five models in four centers. **C** Radar chart comparison of five models in four centers. VFMGL Vision Foundation Model General Lightweight, EC Endometrial Cancer, AUC Area Under the Curve, TPR True Positive Rate, FPR False Positive Rate, PPV Positive Predictive Value, NPV Negative Predictive Value. Source data are provided as a Source Data file.

pathology scanning equipment, WSIs obtained from different hospitals can vary obviously (Supplementary Fig. 2), posing a challenge of Data Heterogeneity (DH) for AI-based cross-center diagnosis.

In the second use case, we tested and discussed whether VFMGL could be used to identify breast cancer metastases in lymph node slides based on the Camelyon17 dataset[3]. To facilitate comparison with other state-of-the-art algorithms, we trained and validated VFMGL on the original distribution of this dataset. VFMGL achieved prediction accuracies of 0.9889 (11756/11888), 0.9728 (6791/6981), 0.9913 (16863/17011), 0.9708 (25209/25968), and 0.9884 (29004/29345) at various centers, with corresponding AUC values of 0.9992, 0.9973, 0.9995, 0.9977, and 0.9993, respectively, outperforming the comparative algorithms (Fig. 3). Center B had an obviously smaller sample size compared to the other centers, leading to substantial performance degradation of all comparative algorithms. In contrast, VFMGL exhibited more stable predictive performance across all centers without considerable performance degradation at any specific center. VFMGL achieved a high tissue image identification rate of 97.15% (44298/45596) for breast cancer metastases and 99.4% (45325/45597) for non-metastatic breast tissue images, facilitating accurate qualitative assessment of whole-slide images to understand the status of breast cancer cell metastasis in patients. In addition, we also performed a performance comparison between VFMGL and the Logit-based KD method (Supplementary Fig. 12). More details of the results can be found in Supplementary Tables 2 and 16.

## Segmentation of prostate in MRI

Prostate ailments (e.g., prostate cancer, prostatitis, and benign prostatic hyperplasia) are prevalent conditions among males[4,31,38]. Precise delineation of the prostate gland from magnetic resonance imaging (MRI) scans is essential for diagnosing and strategizing treatment for these ailments. However, due to differences in imaging protocols, the use of endorectal coils, or demographic data, prostate MRI data from different centers exhibit significant inter-center DH, greatly affecting the accuracy of AI segmentation.

In the third use case, we tested and discussed the performance of VFMGL in segmentation tasks in the face of such DH issues using a prostate dataset[4,38]. VFMGL achieved Dice accuracies of 0.9340, 0.9328, 0.9620, 0.9191, 0.9383, and 0.9080 at various centers (Fig. 4). VFMGL had an average Average Symmetric Surface Distance (ASSD) of 5.9683 across centers, with an average sensitivity of 0.9095 and exceptionally high average specificity of 0.9956, demonstrating excellent capability in correctly identifying non-prostate regions. Partial segmentation results of VFMGL on test sets from various centers are shown in Supplementary Fig. 14, illustrating how inter-center heterogeneity, such as differences in imaging parameters leading to variations in brightness, may interfere with deep learning networks. Contrasting methods based on deep learning networks might result in over-segmentation of target areas or mis-segmentation in distant regions due to such interference. More details of the results can be found in Supplementary Table 3.

## Table 1 | Patient information

| Center | Set | Type | Age (mean ± std) | FIGO stage | | | | Histopathologic type | |
|---|---|---|---|---|---|---|---|---|---|
| | | | | I | II | III | IV | I | II |
| Center A | Train-set (410) | NMI (49) | 50.3673 ± 9.4001 | 46 | 0 | 2 | 1 | 42 | 7 |
| | | MI (361) | 55.9335 ± 8.1279 | 261 | 31 | 61 | 8 | 336 | 25 |
| | Test-set (275) | NMI (34) | 47.5588 ± 7.2413 | 34 | 0 | 0 | 0 | 34 | 0 |
| | | MI (241) | 54.8423 ± 7.8782 | 168 | 21 | 50 | 2 | 216 | 25 |
| Center B | Train-set (42) | NMI (8) | 48.3750 ± 5.8294 | 8 | 0 | 0 | 0 | 8 | 0 |
| | | MI (34) | 52.8824 ± 7.9839 | 28 | 2 | 3 | 1 | 32 | 2 |
| | Test-set (29) | NMI (6) | 45.5000 ± 10.4451 | 6 | 0 | 0 | 0 | 6 | 0 |
| | | MI (23) | 55.5652 ± 10.2772 | 16 | 3 | 4 | 0 | 22 | 1 |
| Center C | Train-set (22) | NMI (1) | – | 1 | 0 | 0 | 0 | 0 | 1 |
| | | MI (21) | 60.9524 ± 8.1944 | 15 | 2 | 3 | 1 | 19 | 2 |
| | Test-set (15) | NMI (1) | – | 1 | 0 | 0 | 0 | 1 | 0 |
| | | MI (14) | 60.1429 ± 10.2421 | 9 | 3 | 2 | 0 | 14 | 0 |
| Center D | Train-set (176) | NMI (11) | 51.1818 ± 11.7883 | 11 | 0 | 0 | 0 | 10 | 1 |
| | | MI (165) | 54.6788 ± 8.1195 | 127 | 5 | 31 | 2 | 148 | 17 |
| | Test-set (118) | NMI (8) | 51.0000 ± 3.1168 | 7 | 0 | 1 | 0 | 8 | 0 |
| | | MI (110) | 54.4636 ± 7.8503 | 91 | 3 | 14 | 2 | 101 | 9 |
| External validation center E | Test-set (63) | NMI (1) | – | 1 | 0 | 0 | 0 | 1 | 0 |
| | | MI (62) | 55.0968 ± 8.9511 | 42 | 11 | 7 | 2 | 60 | 2 |
| External validation center F | Test-set (117) | NMI (8) | 54.6250 ± 13.7730 | 8 | 0 | 0 | 0 | 7 | 1 |
| | | MI (109) | 54.9358 ± 8.9134 | 91 | 2 | 15 | 1 | 99 | 10 |

*NMI* Non myometrial invasion, *MI* myometrial invasion, *std* standard deviation

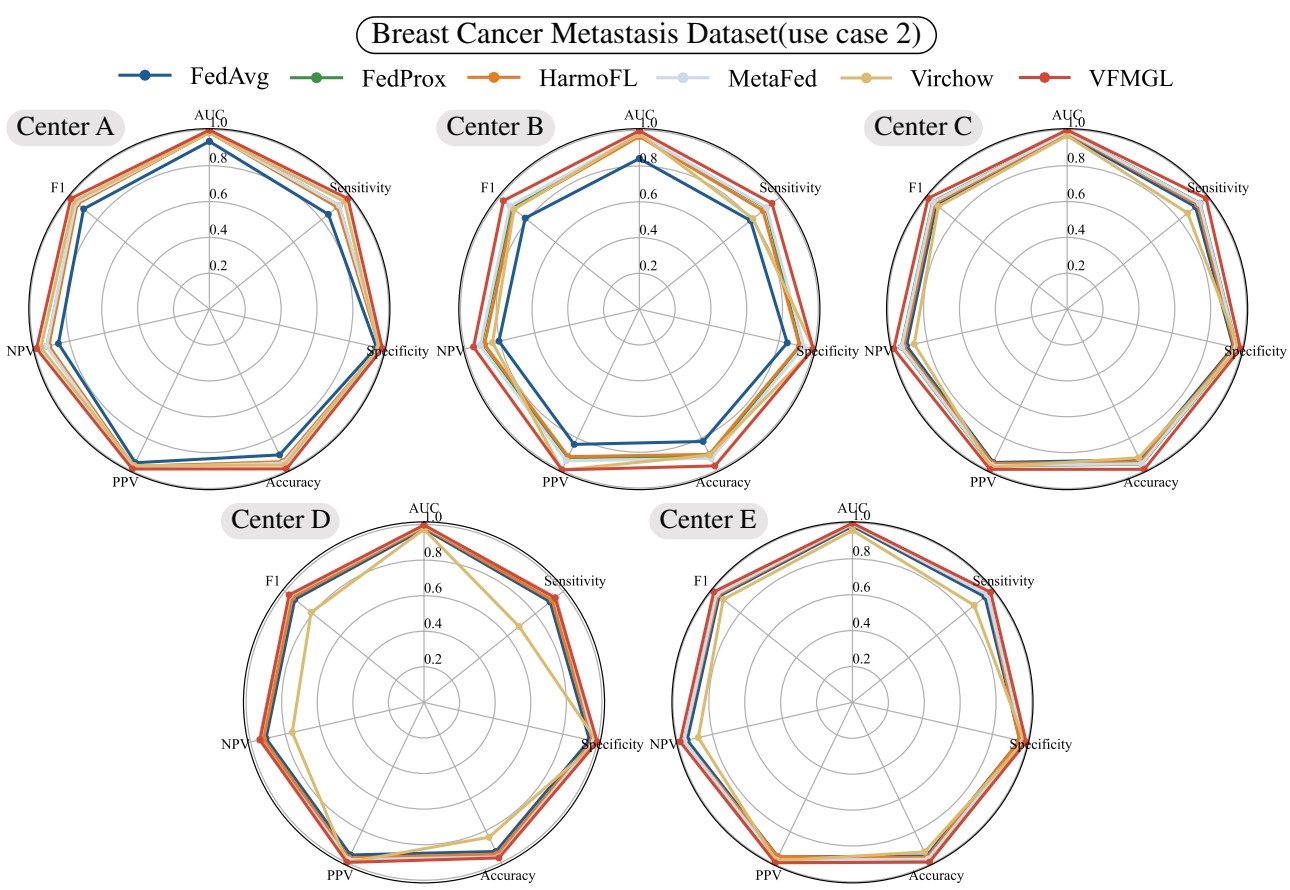

**Fig. 3 | Radar chart comparison of five models in five centers.** VFMGL Vision Foundation Model General Lightweight, AUC Area Under the Curve, PPV Positive Predictive Value, NPV Negative Predictive Value. Source data are provided as a Source Data file.

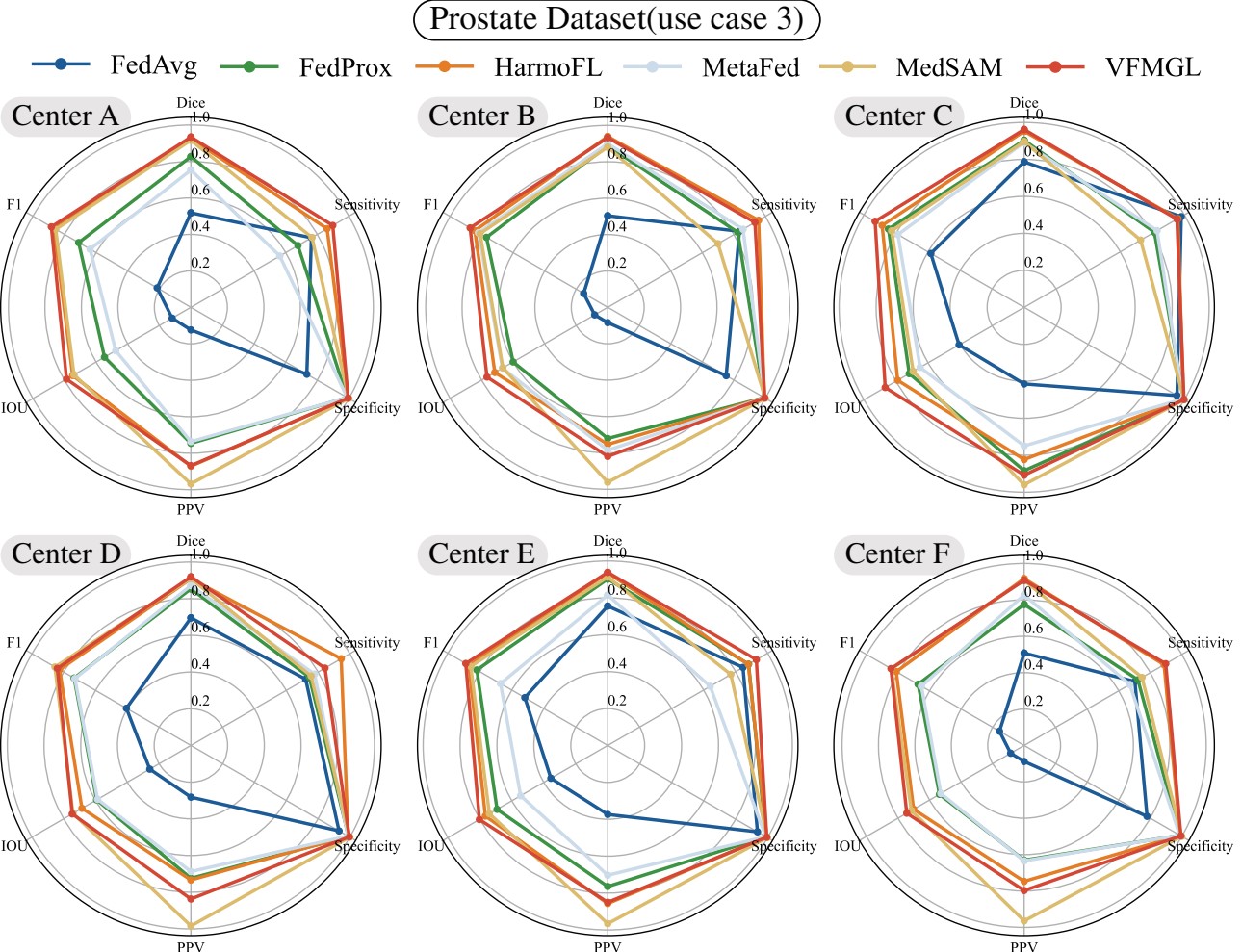

**Fig. 4 | Radar charts for the six centers.** VFMGL Vision Foundation Model General Lightweight, IOU Intersection Over Union, PPV Positive Predictive Value. Source data are provided as a Source Data file.

**VFMGL segments multiple cell nuclei in slices of various organs**

The segmentation of cell nuclei provides fundamental visual information and morphological features such as size, shape, or color. These pieces of information and features not only aid in further processing of pathological images (e.g., classification or tissue segmentation) but also assist pathologists in diagnosing and analyzing the progression of conditions (e.g., cancer diagnosis, assessment, and prognosis)[39–41]. However, the complex background of pathological images and the scattered distribution of cell nuclei greatly increase the difficulty of segmenting cell nuclei.

In the fourth use case, we utilized three public datasets to create a cell nucleus segmentation dataset[40–42]. Due to the dataset's composition of multiple organs and cell types, VFMGL not only faces the DH problem discussed in case 2 but also must address differences in cell nucleus morphology across multiple organs and tissue types. VFMGL achieved Dice accuracies of 0.7509, 0.7658, 0.7735, 0.7410, 0.7568, and 0.7899 at various centers. Compared to other methods, VFMGL demonstrated superior overall performance at each center(Fig. 5). Despite the imbalance in sample numbers across multiple centers, VFMGL did not experience noteworthy performance degradation at specific centers (such as centers E and F), unlike the comparative algorithms. VFMGL achieved an average ASSD of 4.1648, an average sensitivity of 0.5981, and an average specificity of 0.9469 for this task. Partial segmentation results of VFMGL on the test sets of various centers are presented in Supplementary Fig. 15. Staining differences and variations in nuclear morphology significantly impact the

segmentation performance of deep learning models. For example, in the case of center C, changes in staining depth interfere with the deep network's segmentation of non-nucleus areas, while also affecting the segmentation of nuclear contours, resulting in the adhesion of two segmented regions. Compared to other methods, VFMGL more effectively distinguishes adjacent cell nucleus regions and reduces instances of mis-segmentation. More details of the results can be found in Supplementary Table 4.

**VFMGL exhibits robustness**

The performance of deep learning models is influenced by various factors. To further evaluate the robustness of the model, we conducted validation on the EC dataset (use case 1) and public dataset (use case 2–4) from the following two aspects:

Part I: To assess the impact of patient demographic differences on the performance of VFMGL, we grouped the test set data based on the average age (Age = 54.7) of the overall patient population and created two groups (Supplementary Table 5). Group 1 and Group 2 exhibited significant differences in the distribution of positive and negative samples, with Group 2's Centers C and D containing only positive samples. In Group 1 (Age ≤ 54.7), the AUC values for each center were 0.760, 0.840, 0.750, and 0.777, respectively. In Group 2 (Age > 54.7), the AUC values for Center A and Center B were 0.829 and 0.923, respectively. Since NMI samples were not present in Center C and Center D (Group 2), we used the cutoff from the training set to calculate the prediction accuracy for these two centers, which were 0.900

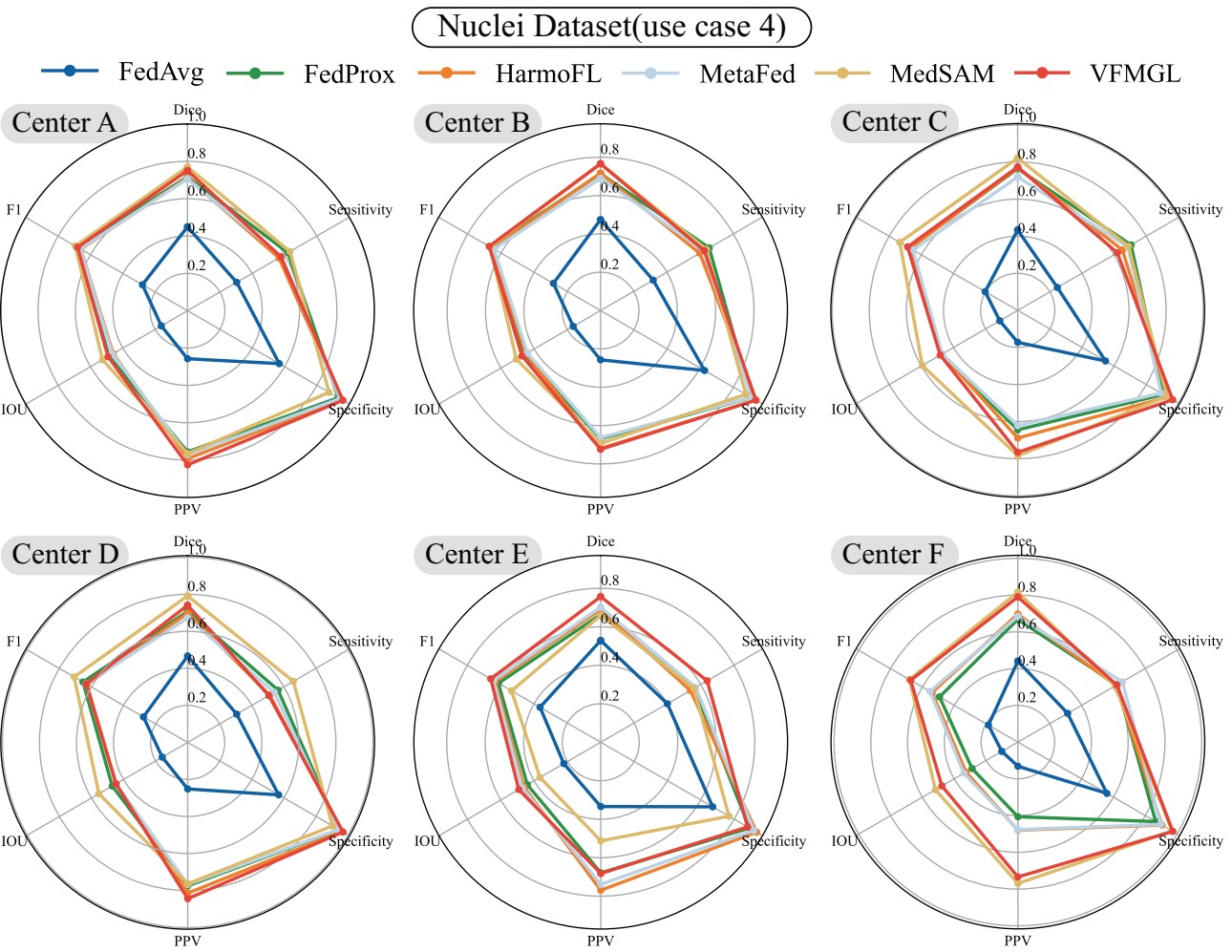

**Fig. 5 | Radar charts for the six centers.** VFMGL Vision Foundation Model General Lightweight, IOU Intersection Over Union, PPV Positive Predictive Value. Source data are provided as a Source Data file.

(cutoff = 0.9333) and 0.836 (cutoff = 0.8767), respectively. The experimental results indicated that VFMGL maintained robust predictive performance even after grouping the data, with particularly high prediction accuracy for Centers C and D in Group 2.

Part II: Considering the impact of data distribution on VFMGL's performance, we conducted random permutation experiments using six different data split ratios and random seeds for dataset creation (Supplementary Tables 6, 9–11). The experimental results (Supplementary Figs. 3, 8–10) demonstrated that VFMGL exhibited strong resistance to variations in data distribution, with only slight fluctuations in predictive performance. In the EC task (use case 1), across the six data distributions, the average AUC performance of VFMGL for the four centers was as follows: $0.778 \pm 0.042$, $0.845 \pm 0.026$, $0.837 \pm 0.019$, and $0.852 \pm 0.022$. The robustness of VFMGL was further validated on three public datasets. In the breast cancer pathology image classification task (use case 2), the AUC performance of VFMGL at each center was as follows: $0.9990 \pm 0.0001$ (center A), $0.9978 \pm 0.0003$ (center B), $0.9988 \pm 0.0004$ (center C), $0.9966 \pm 0.0008$ (center D), and $0.9989 \pm 0.0002$ (center E). In the prostate MRI image segmentation task (use case 3), the Dice performance of VFMGL at each center was: $0.9134 \pm 0.0121$, $0.9258 \pm 0.0098$, $0.9546 \pm 0.0060$, $0.9297 \pm 0.0100$, $0.9246 \pm 0.0106$, and $0.9013 \pm 0.0072$. In the pathology image nucleus segmentation task (use case 4), the Dice performance of VFMGL at each center was: $0.7665 \pm 0.0203$, $0.7652 \pm 0.0105$, $0.7823 \pm 0.0187$,

$0.7710 \pm 0.0160$, $0.7562 \pm 0.0095$, and $0.8233 \pm 0.0209$. These extensive experiments show that VFMGL maintained excellent robustness across various scenarios and tasks, even when facing different variations in data distribution. More detailed results can be found in Supplementary Tables 12–14.

### VFMGL demonstrates cross-center generalization

AI models confront distinct heterogeneity implications across various tasks and datasets. Within the realm of multi-center DH, we conducted additional assessments to substantiate VFMGL's cross-center generalization in the aforementioned four utilization scenarios. As depicted in Fig. 6, for the EC-MI task (Fig. 6a), Model A demonstrates cross-center generalization. The models from centers B and D generalize between these two centers, while the models from centers C and D exhibit generalization between these two centers. For breast cancer histopathological image classification (Fig. 6b), there is mutual generalization among centers A, B, D, and E, and model C exhibits generalization across all centers. In the prostate MRI segmentation task (Fig. 6c), VFMGL demonstrates mutual generalization between centers D, E, and F. Model A exhibits generalization across centers B, C, D, and E, and models D, E, and F demonstrate generalization on center B. In the histological cell nucleus segmentation task (Fig. 6d), mutual generalization is observed among Centers A, B, C, and D, as well as between centers E and F.

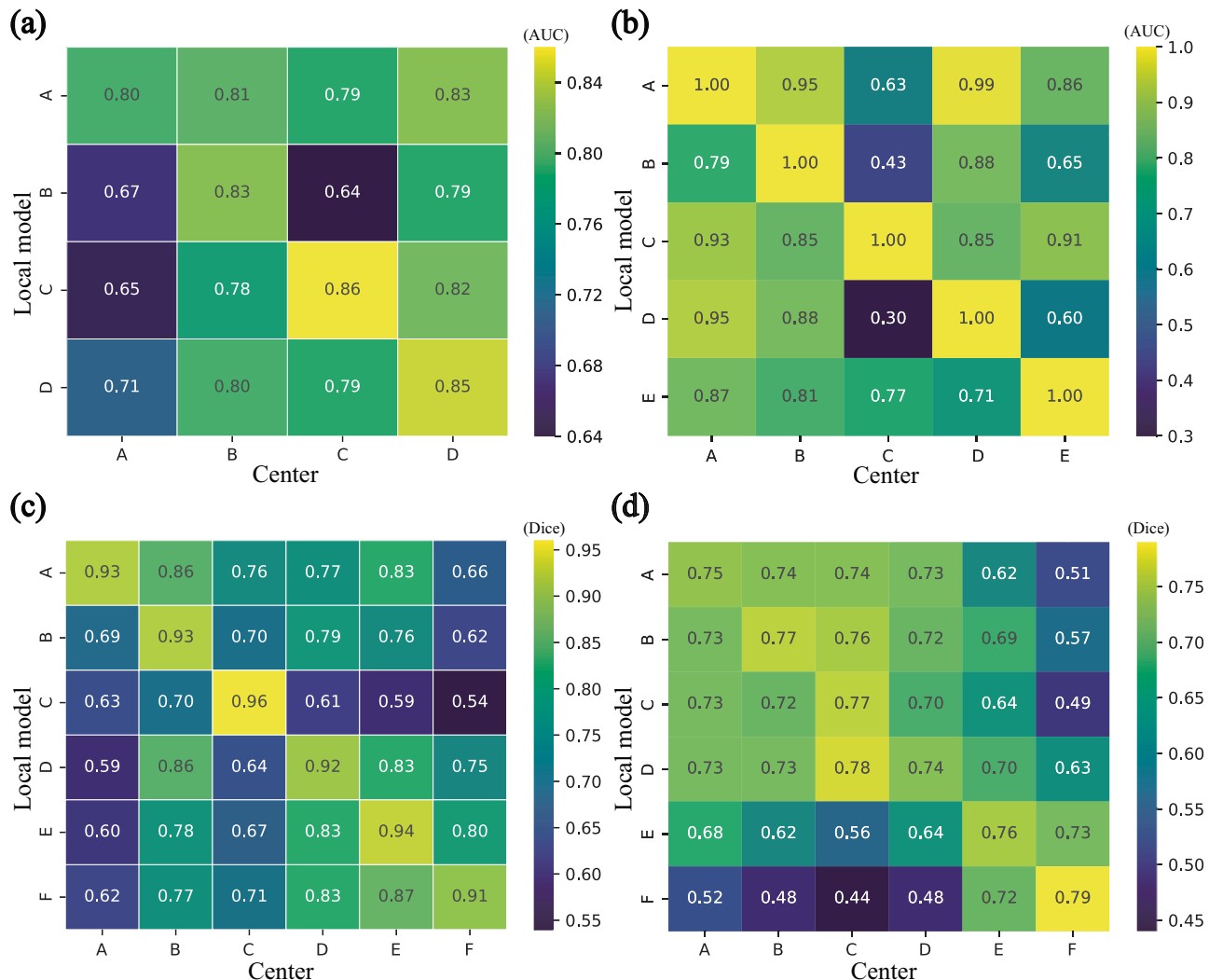

**Fig. 6 | Cross-center generalization of the VFMGL on the four tasks. a** Cross-center generalization maps for use case 1. **b** Cross-center generalization maps for use case 2. **c** Cross-center generalization maps for use case 3. **d** Cross-center generalization maps for use case 4. AUC Area Under the Curve. Source data are provided as a Source Data file.

## VFMGL-based predictions are explainable

To investigate the interpretability and diagnostic foundation of VFMGL in diagnosing the presence or absence of MI in EC (use case 1), we conducted Class Activation Mapping visualizations[43]. In Fig. 7a, we present eight visualization results from four centers, including four cases with MI and four cases with Non-MI (NMI). The images demonstrate that, for MI cases, VFMGL exhibits strong activation responses in the region of the EC lesion (red region), while for NMI cases, VFMGL shows low activation responses in the uterine region (blue region).

To further explore the representational capabilities of VFMGL on multi-center data, we conducted Principal Component Analysis (PCA)[44] visualization of features from each center. Figure 7b displays the features learned by VFMGL at each center, indicating the consistency in multi-center distribution and the effectiveness of discriminative classification features. Additionally, Fig. 7c illustrates the overall distribution of prediction scores for both classes. The results indicate that there are significant differences in the positive and negative sample prediction scores of VFMGL at centers A, B, and D ($p < 0.05$). The predictive distributions of VFMGL model on the training and testing sets are consistent.

## Adaptive knowledge and common knowledge learned by VFMGL

To further explore the interpretability of VFMGL, we analyzed the adaptive knowledge and common knowledge learned by VFMGL on the EC dataset (use case 1). Adaptive knowledge refers to the unique classification feature characteristics within each center, while common knowledge refers to the classification feature characteristics shared by all centers[45]. Based on the construction process of VFMGL, we visualize the correlation heatmaps of the model features from both stages, reflecting the intra-center feature relationships and cross-center feature relationships.

In Phase I, VFMGL constructs robustness critical layers adaptively based on the common knowledge from VFM. The details of the robustness critical layers of local models in each center are shown in Supplementary Tables 7 and 15. Taking the features learned by VFMGL on MI cases as an example, in the analysis of adaptive knowledge, the features of each center exhibit strong intra-group correlation (Fig. 8a and c), while inter-group correlation is weak, especially between Center A and Center D. The heatmap of common features (Fig. 8b and d) indicates good inter-group correlation of common features. The heatmaps of the training set and test set

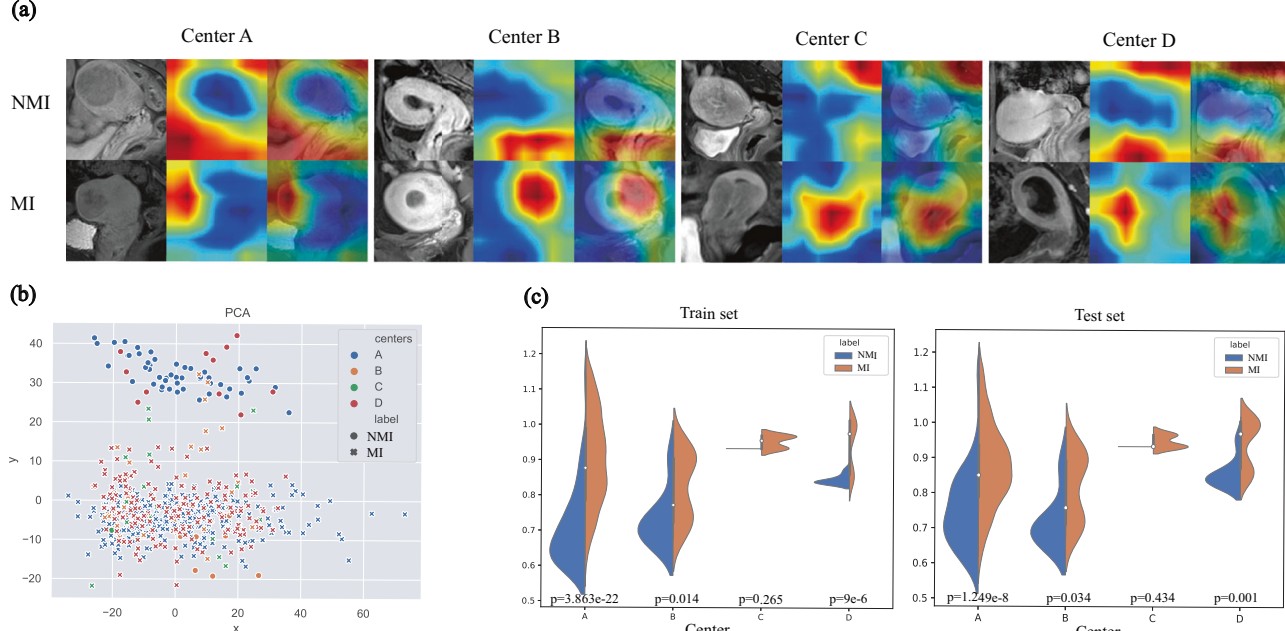

**Fig. 7 | Visualizations for model interpretability. a** The heatmap shows the information acquired by the VFMGL for images in the MI and NMI classes. The red areas indicate a high level of model attention, while the blue areas indicate a low level of model attention. **b** The feature distribution of VFMGL in the four centers. **c** The score charts illustrate the MI and NMI cases of the four centers evaluated by the VFMGL. The statistical test used for this data analysis is the Independent *t*-test (two-tailed). In the training set, Center A ($n = 410$, $p = 3.863e-22$; mean ± std: 0.707 ± 0.109, 0.904 ± 0.128); Center B ($n = 42$, $p = 0.014$; mean ± std: 0.731 ± 0.078,

0.823 ± 0.093), Center C ($n = 22$, $p = 0.265$; mean ± std: -, 0.953 ± 0.017), Center D ($n = 176$, $p = 9e-6$; mean ± std: 0.846 ± 0.013, 0.943 ± 0.070); In the test set, Center A ($n = 275$, $p = 1.249e-8$; mean ± std: 0.753 ± 0.107, 0.875 ± 0.115); Center B ($n = 29$, $p = 0.034$; mean ± std: 0.721 ± 0.079, 0.815 ± 0.094), Center C ($n = 15$, $p = 0.434$; mean ± std: -, 0.948 ± 0.017), Center D ($n = 118$, $p = 0.001$; mean ± std: 0.858 ± 0.041, 0.940 ± 0.069). MI Myometrial Invasion, NMI Non Myometrial Invasion, PCA Principal Component Analysis, p significance value, std standard deviation. Source data are provided as a Source Data file.

(Fig. 8a, b vs Fig. 8c, d) show consistent shapes in corresponding positions, indicating the stability of the features learned and the relationships between them in this phase. Further comparison of the correlation heatmaps of MI (Fig. 8a–d) and NMI (Supplementary Fig. 7a–d) shows that adaptive features from each center consistently appear in both types of heatmaps, demonstrating the potential value of enhancing the robustness of VFMGL predictions.

In Phase II, VFMGL utilizes common knowledge from the shared model to enhance generalization performance. Compared to Phase I, the features of MI cases in each center do not exhibit strong intra-group correlation (Fig. 8e, g); however, in the heatmap of common features (Fig. 8f, h), the common features learned by VFMGL in this phase show high inter-group correlation and maintain weak intra-group correlation. Common features possess similar representation capabilities across centers, assisting VFMGL in achieving generalization performance on multi-center data. The correlation heatmap of adaptive features and common features for NMI cases can be seen in Supplementary Fig. 4.

**The lightweight performance of VFMGL**

We evaluate the performance of the proposed VFMGL technique using model parameter count and Floating Point Operations (FLOPs). Parameter count refers to the total number of parameters that need to be trained during model training, measuring the model's spatial complexity. FLOPs represent the number of floating-point operations during model inference, i.e., the theoretical computational load of the model, measuring the model's computational time complexity. In our experiments, the parameter count of the VFM is approximately 86 M, with a computational load of about 21.96 GFLOPs, while the parameter count of the VFMGL is approximately 11 M, with a computational load of about 1.82 GFLOPs. After lightweighting using the HAKD technique,

the compression ratio of the model parameter count is approximately 1:8, and the computational load is compressed by a factor of 1:12. Deployment and use of the VFMGL require less than 500MiB of GPU memory or CPU usage, while maintaining fast inference speeds.

**Ablation study**

We further conducted ablation experiments on public datasets (use case 3-4) to investigate the impact of HGKT and DDBL on model performance. Figure 9 and Supplementary Fig. 16 illustrate the performance changes of VFMGL. Subfigures (a), (b), and (c) respectively represent the model performance of using only HGKT, HGKT + KD, and HGKT + KD + DDBL. To facilitate the comparison of performance differences among these methods, we further plotted the performance variations. Subfigures (d), (e), and (f) correspond to the performance changes from subfigure (a) to (b), subfigure (a) to (c), and subfigure (b) to (c), respectively.

In use case 3, KD improved the model's average performance across centers by 0.0110, while KD + DDBL raised the average performance by 0.0429. Among the 36 cross-center generalization results, KD + DDBL significantly enhanced the model's performance in 22 results (Fig. 9e). In use case 4, KD improved the average performance across centers by 0.0068, while KD + DDBL increased it by 0.0097. For the 36 cross-center generalization results, KD + DDBL significantly enhanced the model's performance in 25 results (Supplementary Fig. 16e). These experimental results indicate that the HGKT method enables local models to achieve excellent generalization performance, while KD and DDBL further enhance the overall performance of VFMGL across all centers. These comparisons show that, without compromising local models' performance on their own data, their performance on data from other centers can still be improved.

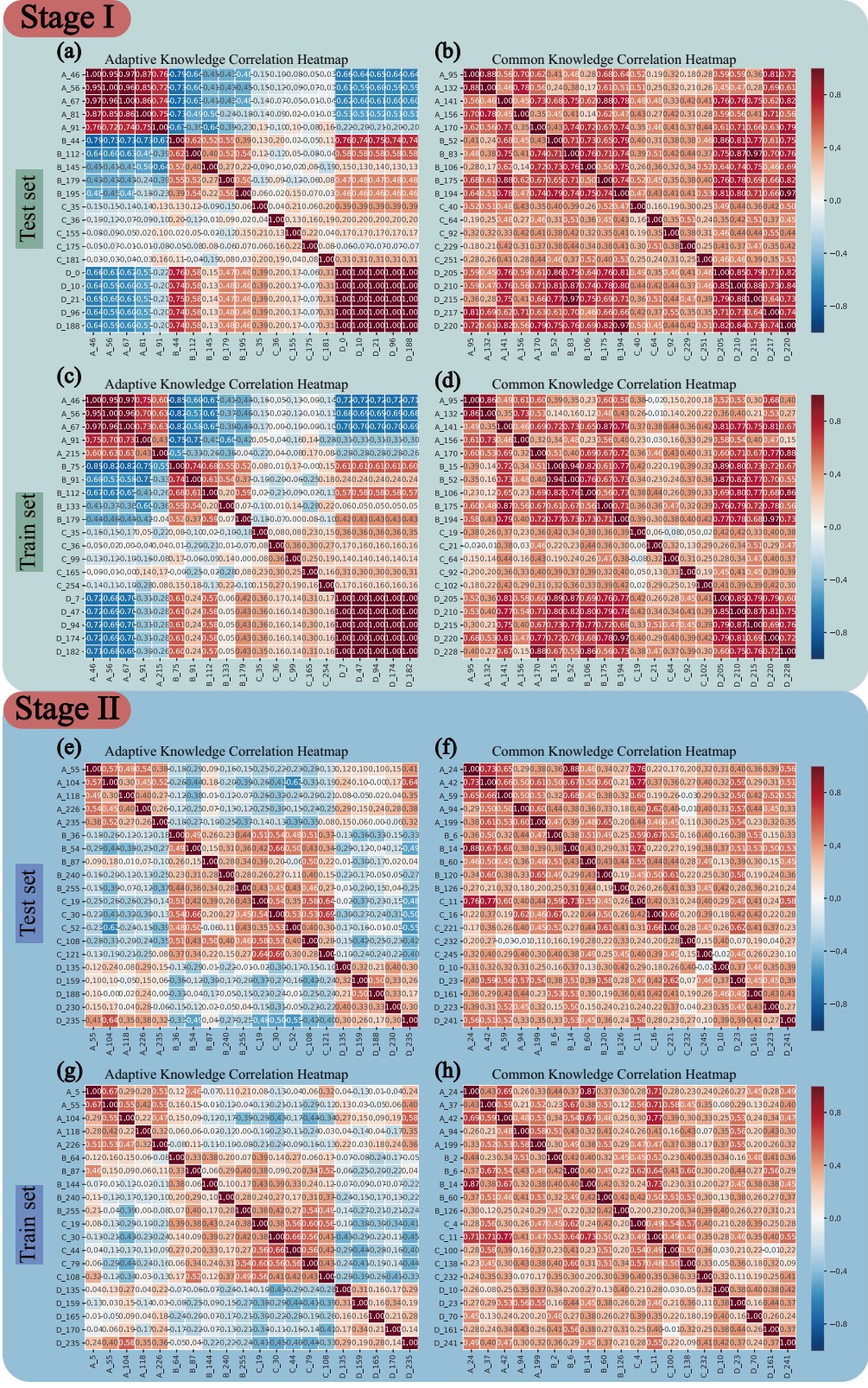

**Fig. 8 | Correlation heatmaps of adaptive knowledge and common knowledge.** **a** Adaptive knowledge correlation heatmap of the first stage of VFMGL in the test set. **b** Common knowledge correlation heatmap of the first stage of VFMGL in the test set. **c** Adaptive knowledge correlation heatmap of the first stage of VFMGL in the training set. **d** Common knowledge correlation heatmap of the first stage of VFMGL in the training set. **e** Adaptive knowledge correlation heatmap of the second stage of VFMGL in the test set. **f** Common knowledge correlation heatmap of the

second stage of VFMGL in the test set. **g** Adaptive knowledge correlation heatmap of the second stage of VFMGL in the training set. **h** Common knowledge correlation heatmap of the second stage of VFMGL in the training set. The first column shows the heatmap of correlation for adaptive features, while the second column displays the heatmap of correlation for common features. VFMGL Vision Foundation Model General Lightweight. Source data are provided as a Source Data file.

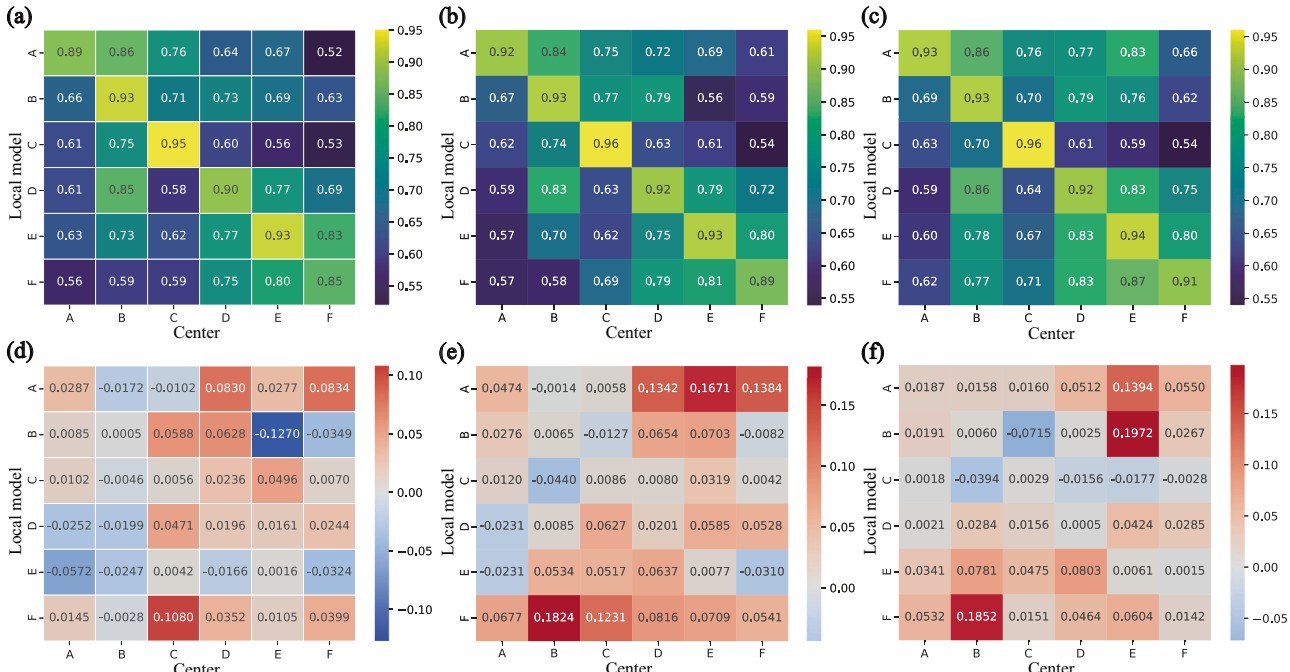

**Fig. 9 | Ablation experiments of VFMGL in use case 3. a** Performance when constructing models using only the HGKT method. **b** Performance when constructing models using only the HGKT + KD method. **c** Performance when constructing models using the HGKT + KD + DDBL method. **d** Models performance change from subfigure (**a**) to (**b**). **e** Models performance change from subfigure (**a**) to (**c**). **f** Models performance change from subfigure (**b**) to (**c**). HGKT Heterogeneous-model General Knowledge Transfer, KD Knowledge Distillation, DDBL Data Deduction in Batch Level. Source data are provided as a Source Data file.

## Discussion

Precision diagnosis based on medical imaging (MRI, WSI, etc.) can provide patients with more personalized medical services, reduce treatment risks, and improve treatment outcomes[1–7]. However, owing to the intricate nature of imaging manifestations in many diseases, achieving accurate diagnoses remains challenging[2,46]. Deep learning (DL) is emerging as a potential solution to address this issue. DL can directly extract target information from images, perform quantitative analysis, and provide objective reference information for diagnosis. Tang et al. employed deep learning (DL) to identify specific neuropathological lesions in immunohistochemistry-stained tissue slices[47]. Huang et al. developed PENet for detecting pulmonary embolism[48]. Jiang et al. utilized DL to predict peritoneal recurrence of gastric cancer on CT images[49].

However, due to the challenges of multi-center DH[4,11,12], the aforementioned DL systems may face performance limitations. Because of the need for medical data privacy and security, medical centers generally prohibit the sharing of patient data, making it difficult for single centers to obtain diverse, large-scale medical data for research on DL diagnostic systems. A large number of medical imaging studies are based on training DL models with single-center data[50–53], using standardized machines and imaging protocols for data collection. However, in scenarios involving multiple centers, institutions, and operators, traditional DL training methods often encounter challenges related to DH. These mainly include: (1) DL models trained in a single center may not work in other centers due to differences in image acquisition equipment or parameters across centers; (2) DL models may lack robustness when facing prediction challenges such as differences in patient demographics; Additionally, (3) the healthcare sector has strict regulations and ethical requirements regarding the transparency and interpretability of model decisions.

The powerful feature extraction capability and general representation ability of VFMs endow them with robustness in predicting unknown objects. However, training VFMs for medical purposes requires vast amounts of data, abundant computational resources, and

extensive time. Additionally, conducting interpretability work for foundation models is challenging[8]. In this study, we proposed a HGKT technique for lightweight parameterization of foundation model parameters, adapting to utilize the universal knowledge from VFM to assist deep learning models in building robust critical layers, enabling DL models to possess good robustness for tasks within the center. The preliminary lightweighting results show that the model parameter size can be compressed to one-eighth of its original size, and the theoretical computational workload can be reduced to one-twelfth of its original value. Robustness testing(use case 1–4) validated the performance of VFMGL, showing predictive stability under various data distribution. For example, in use case 1, AUC variations were observed in robustness testing: $0.778 \pm 0.042$, $0.845 \pm 0.026$, $0.837 \pm 0.019$, and $0.852 \pm 0.022$. Further exploration of features learned by local models in this phase revealed unique characteristics distinguishing them from other centers, with stable feature expression and relationship structures in local datasets. General knowledge from VFM helped local models achieve robust feature expression on local data, enhancing their robustness against perturbations such as population differences and data distribution changes.

In the VFMGL framework, each center refrains from sharing medical data and collaborates solely by sharing model parameters to train models, ensuring the privacy and security of patient data. We propose a DDBL method tailored to multi-center DH, which selects data with low heterogeneity from each center's dataset for training local models (Supplementary Fig. 11). Leveraging knowledge distillation techniques, we aim to assist local models in acquiring common knowledge from the shared model, thereby enhancing the generalization performance of local models. Compared to existing FL algorithms, the developed VFMGL demonstrated superior performance, proving its effectiveness in medical classification and segmentation(use case 1–4). Using local models from VFMGL for testing on data from other centers demonstrated that VFMGL possesses cross-center generalization capabilities(use case 1–4). The features learned by the local models at this stage

exhibit sharing properties, and these features demonstrate representational capabilities across multiple centers.

The capability of VFMGL to handle various medical tasks has been validated on one private dataset (use case 1) and three public datasets (use cases 2–4). In use case 1, for well-differentiated endometrioid endometrial carcinoma (EEC) patients, there is a potential to preserve fertility if myometrial invasion is absent[2]. The combined test sets from centers A-E comprised 157 cases of well-differentiated EEC patients (including an external test set), with VFMGL achieving a prediction accuracy of 70.70% (111/157) for the presence or absence of MI in such patients. This aids in identifying individuals without MI, where the lesion is confined to the uterus, potentially enabling fertility-sparing treatments. Furthermore, VFMGL achieved promising results in breast cancer histological image classification (use case 2), histological cell nucleus segmentation (use case 3), and prostate MRI segmentation (use case 4), demonstrating VFMGL's potential in medical image classification and segmentation.

In addition to these advantages, the predictive performance of VFMGL exhibits excellent interpretability. VFMGL can effectively identify adaptive features, demonstrating high similarity within each data center and low similarity between data centers. It maintains stable representation relationships within local data, promoting the differentiation of positive and negative samples, enhancing predictive stability. VFMGL reveals highly similar crucial common features between each data center, allowing for accurate differentiation of EC with and without MI across different data centers. An overall analysis of the multi-center data features learned by VFMGL indicates that the distribution of features has better consistency while possessing excellent classification characteristics.

Furthermore, considering the potential future real-world application of VFMGL, where centers may lack independent diagnostic capabilities or independent centers not involved in VFMGL training may need to use the VFMGL for disease diagnosis, we analyzed the similarity between the data features from independent centers and the data features of VFMGL training set data based on the idea of data inference. This allowed us to select the better local model for independent external validation testing, and experimental results (use case 1) demonstrated that the higher the data features similarity (Center A: 0.15; Center B: −0.05; Center C: −0.06; Center D: 0.08), the better the predictive performance of the local model for that center (Model A was 0.742; Model B: 0.694; Model C: 0.661; Model D: 0.710).

A caveat of our observations is that the ground truth might not be perfect. Firstly, this study did not explore the potential relationship between common and adaptive features and clinical information, which would enhance the interpretability of the VFMGL mechanism and provide higher clinical value. Secondly, this study primarily investigated model robustness and generalization without exhaustively optimizing each model hyperparameter, suggesting that VFMGL's performance might be better than described in the paper, and considering the speed of knowledge distillation, we initially chose medium-sized VFMs for experimentation. In fact, the HAKD method can be used to distill larger-scale VFMs. Finally, the contributions of adaptive features and common features to model robustness and generalization still require further exploration, aiding in the discovery of methods to further enhance model performance.

## Methods
### Materials and pre-processing
EC dataset (use case 1): This study was implemented under the approval of the Jiangmen Central Hospital, the Yuebei People's Hospital, Affiliated Dongguan Hospital Southern Medical University, the Maoming People's Hospital, the Kaiping Central Hospital and the Third Affiliated Hospital of Guangzhou Medical University, and conducted in accordance with the 1964 Helsinki Declaration and its later amendments or comparable ethical standards. Informed consent was waived

by our Institutional Review Board because of the retrospective nature of our study. For Center A ($n = 685$), we included patient data who underwent total hysterectomy for endometrial cancer from August 2010 to December 2022. For Centers B ($n = 71$), C ($n = 37$), D ($n = 294$), and E ($n = 63$), we included patient data who underwent total hysterectomy for endometrial cancer from December 2016 to February 2023. In addition, we have collected 117 cases of patients from a new center F. In total, data from 6 medical centers comprising 1267 patients were included. Inclusion criteria were: (1) histologically confirmed endometrial cancer (malignant epithelial tumors of uterus); (2) underwent total hysterectomy; (3) had pelvic MRI images within 21 days before surgery; (4) had complete postoperative pathological results. Exclusion criteria were: (1) interval between pelvic MRI examination date and surgery date exceeding 21 days; (2) received neoadjuvant therapy before surgery; (3) presence of artifacts or poor image quality in pelvic MRI; (4) concomitant presence of other malignant tumors, such as ovarian cancer, cervical cancer, etc. The dataset was randomly divided into training and testing sets at a ratio of 6:4 for each center, and the distribution of patient data is shown in Table 1 and Supplementary Fig. 5a. During the initial phase of the study, highly experienced radiologists delineated the Region of Interest (ROI) as the input for the local models. This ROI constitutes a rectangular region encompassing the entirety of the uterus. All ROI images have been resized to $224 \times 224 \times 3$. Further information on ROI acquisition is available in Supplementary Note 1.

Breast cancer histology image dataset (use case 2): The dataset[12] is derived from the CAMELYON17 dataset[3], which includes patient slides from five different medical centers in the Netherlands. These slides comprise both H&E slides and IHC slides. The dataset consists of 450,000 patches of breast cancer metastases on lymph node slides, all patches being of size $96 \times 96 \times 3$. For each medical center (Supplementary Table 8 and Supplementary Fig. 5b), 20% of the data is allocated to the test set, while the remaining 80% is divided into a 4:1 ratio for the training and validation sets.

Prostate MRI dataset (use case 3): The dataset[4,38] comprises T2-weighted MRI data from six different data centers of three public datasets[54–56], with images resized to $384 \times 384$. Different centers utilized various scanners, field strengths, resolutions and coil type. Center A employed a GE scanner with a field strength of 3.0 T and a resolution of 0.25/2.2–3 mm, using endorectal coil. Center B utilized a Siemens scanner with a field strength of 1.5 T and a resolution of 0.625/3.6 mm, also using endorectal coil. Center C used a Siemens scanner with a field strength of 3.0 T and a resolution of 0.67–0.79/1.25 mm, without using endorectal coil. Center D employed a Siemens scanner with a field strength of 3.0 T and a resolution of 0.6–0.625/3.6–4 mm, using endorectal surface coil. Center E utilized a Philips scanner with a field strength of 1.5 T and a resolution of 0.4/3 mm, using endorectal coil. Center F employed a Siemens scanner with field strengths of 1.5 T and 3.0 T, with a resolution of 0.325–0.625/3–3.6 mm, without using endorectal coil. For each medical center, 20% of the data is allocated to the test set, while the remaining 80% is divided into a 4:1 ratio for the training and validation sets (Supplementary Table 8 and Supplementary Fig. 5c).

Histology nuclei dataset (use case 4): The dataset consists of three public datasets: MoNuSAC2018[42], MoNuSAC2020[40], and TNBC[41]. For the data from MoNuSAC2020, it is divided into four centers based on different hospitals, forming a total of six centers. The division criterion follows the official multi-organ split, where each organ group includes specific hospitals without overlapping with other groups. The dataset includes epithelial cells, lymphocytes, neutrophils, macrophages, normal epithelial cells, myoepithelial breast cells (located in ducts and lobules), invasive carcinoma cells, and more. These cells are sourced from various organs such as breast, kidney, lung, prostate, and others. All images have been resized to $256 \times 256 \times 3$. For each medical center, 20% of the data is allocated to the test set, while the remaining 80% is

divided into a 4:1 ratio for the training and validation sets (Supplementary Table 8 and Supplementary Fig. 5d).

Experimental Setting: We use DINOv2[15] as the open-source visual foundation model (VFM) and adopt two commonly used deep learning frameworks, ResNet18[57] and UNet[58], as local and shared models for classification tasks (use case 1–2) and segmentation tasks (use case 3–4), respectively. In use case 1, we use standard cross-entropy loss and an SGD optimizer to update the local model. The model is trained for 20 epochs with a learning rate of 0.0001, weight decay of 0.0001, and a batch size of 25. In use case 2, we use standard cross-entropy loss and an SGD optimizer to update the local model. The model is trained for 10 epochs with a learning rate of 0.01, weight decay of 0.001, and a batch size of 32. In use case 3, we use Dice loss and an Adam optimizer to update the local model. The model is trained for 100 epochs with a learning rate of 0.0001, weight decay of 0.0001, and a batch size of 3. In use case 4, we use Dice loss and an SGD optimizer to update the local model. The model is trained for 100 epochs with a learning rate of 0.001, weight decay of 0.0001, and a batch size of 2. For all four tasks, the optimizer momentum is set to 0.9, and the meta-network parameters are updated using an SGD optimizer with a learning rate and weight decay of 0.001. Local model training is performed once per communication round.

## Vision foundation model general lightweight framework

The VFMGL framework allows each center to train models in a decentralized manner based on VFM general knowledge and shared model knowledge, ensuring both robustness and generalization of the models. It offers an HGKT method to transfer general knowledge from VFM to lightweight models, assisting local models in building robust critical model layers. In the case of multi-center DH, it provides a DDBL method based on shared model knowledge to select low-heterogeneity data from each center's dataset for knowledge distillation from the shared model to the local model, enhancing the generalization of local models with common knowledge.

## The lightweighting of VFM based on HGKT

Unlike convolutional neural networks (CNN) such as ResNet18[57] and VGG16[59], the VFM with a transformer structure possesses robust general visual representation capabilities[15]. This ability allows VFM to maintain outstanding recognition stability across various types of images, including different categories, resolutions, and perturbations. However, this capability comes at the cost of having a large number of model parameters, leading to challenges such as difficult training, deployment, and slow inference speed, especially in low-resource scenarios. Current research primarily employs transfer learning methods[24,60–65] like fine-tune to retrain open-source large models. However, fine-tuning requires a great amount of annotated data. Some researchers propose knowledge distillation methods to leverage large models to guide the training of smaller models[66–69]. However, they face challenges such as label sensitivity, model structure, selection of model layers, and knowledge redundancy.

Given the significant domain differences between natural and medical images, arbitrarily matching model layers for knowledge transfer may not benefit, and could even hinder, medical tasks. Some researchers rely on empirical or experimental selection of fixed intermediate layers for matching and knowledge transfer[35,70,71], a time-consuming task, particularly for VFMs with large model layers. With task and dataset changes, the effectiveness of layer-matching based on manual methods remains uncertain. Moreover, in a multi-center scenario, DH can lead to variation in the knowledge needed for each center's local model[12]. Accordingly, the type of knowledge each center requires from VFMs can vary.

Therefore, we propose a method called HGKT to assist CNNs in learning general knowledge from VFMs and achieve the lightweighting of foundational models. The training tasks and data of open-source

VFMs are typically different from the target domain, implying that not all VFM knowledge is beneficial for the target task. In the process of knowledge transfer, we construct a robust feature transfer network (Supplementary Note 2) that automatically calculates knowledge transfer weights to match the transfer positions of model layers between heterogeneous models and build knowledge transfer channels, helping CNN networks obtain beneficial general knowledge from VFMs for the target task. The robust feature transfer network dynamically updates transfer weights (Supplementary Note 3) based on the performance changes of CNN networks on the target task, adaptively selecting stable representations of general knowledge from VFMs for CNN tasks.

## Building robustness critical layers based on HGKT

Confronted with the impact of multi-center DH on the predictive robustness of AI models and inspired by the universal visual representation capabilities of VFM, we employ HGKT to extract universal knowledge from VFM for constructing the model robustness critical layers. In this study, each medical center possesses its own private data and local model, and the learning process of the first stage of VFMGL is illustrated in Fig. 1b. Utilizing HGKT, an adaptive transfer pipeline is formed from VFMs to local model. Local model autonomously build model robustness based on private data and the general knowledge from VFMs.

CNN parameters exhibit redundancy, and existing studies have demonstrated the possibility of simultaneously ensuring robustness and generalization[72]. The robust feature transfer network calculates the adaptability of model layer transfer between VFMs and local models, reflecting the quantity of general knowledge learned by local model layers from VFMs and the criticality of model layer robustness. We freeze the critical layers of model robustness to prevent local models from forgetting robust knowledge during the learning process of the second stage.

## Enhancing model generalization based on DDBL

Applying deep learning models in a clinical context, the generalization performance of deep learning models will expand the prospects of model applications, especially in terms of cross-center generalization capabilities. The introduction of federated learning (FL)[28] provides a solution for exchanging model knowledge among various medical centers. ProxyFL[29] validated the feasibility of efficient communication and privacy protection in federated learning (FL). In the FL framework, the shared model is aggregated from local models of multiple centers, integrating knowledge and decision information from various sources[12,28,73].

The imaging data from multiple medical centers exhibit DH due to differences in imaging equipment, scanning parameters, and image quality, resulting in biased data features across centers. This allows local models to easily fit local data but perform poorly on data from other centers. As neural networks become increasingly powerful, they can learn specific feature patterns on particular datasets to achieve good performance[26,27]. Therefore, we propose the DDBL method, which selects low-heterogeneity data from each center based on the knowledge of the shared model. This allows local models to learn common knowledge, thereby suppressing the learning of specific feature patterns and leveraging the model's redundant parameters to further enhance cross-center generalization ability.

Local models learn unique feature distributions and biased decision boundaries from private data within their respective centers, with their focus centered on these representative heterogeneous data. This results in better performance on private data but leads to failure when applied to data from other centers[11]. In the second stage of VFMGL (Fig. 1c), both the shared model and local models compute features for randomly sampled batches of private data, with the shared model's classification head generating prediction distributions for each. The

representation knowledge learned by local models on private data facilitates the identification of prominently heterogeneous data, rapidly reducing the discrepancy between model prediction distributions and true label distributions. This enlarges the difference in prediction distributions between the shared model and local models. By measuring the speed of these changes, data with commonalities can be distilled from private data, aiding local models in learning generalized decision boundaries from the shared model, capturing the commonalities of data distributions, and enhancing generalization abilities (Supplementary Notes 4, 5 and 7). After constructing local models, each user sends their local model to the server (Fig. 1a), where the server aggregates shared model parameters using weighted averages[12,28,34].

### Exploration of adaptive knowledge and common knowledge

In order to explore the inference basis of robust models across centers, identify common features for category prediction, and adaptive features for handling differences in data from various research centers, this study utilized VFMGL to extract DL features from each research center. Specifically, local models from each center were employed to extract four sets of DL features from all data samples (use case 1). The mRMR algorithm was applied to filter out 256 most valuable radiomics features from each feature set. Subsequently, the correlation between the four sets of DL features was calculated. Features showing the highest correlation within each data center and between different data centers were determined as adaptive features and common features, respectively. The Pearson correlation coefficient[74] was used in this study, and more details can be found in Supplementary Note 6.

To assess the commonality and adaptability features in VFMGL, a classification heatmap was generated to provide a visual representation of VFMGL's focus on two types of image data from different centers. The correlation heatmap offers insights into how the model emphasizes common and adaptive features within the data. Furthermore, the similarity between adaptive features and common features was evaluated by computing the correlation matrix of DL features from different centers. This allowed an understanding of the relationships and similarities among features from different centers. Through these methods, the study assessed common and adaptive features in VFMGL, revealing VFMGL's attention and relationships to different types of features and data from various centers.

### Statistical analyses

We use AUC, sensitivity, specificity, accuracy, positive predictive value (PPV), and negative predictive value (NPV) to evaluate the performance of VFMGL in classification tasks. Dice coefficient, ASSD, Intersection over Union (IOU), PPV, specificity, and sensitivity are used to evaluate the model's performance in segmentation tasks. These evaluation metrics provided a comprehensive assessment of the predictive capabilities of the algorithms and allowed for statistical comparisons between different models. The ROC curve was used to illustrate the overall performance of the different modelling methods, and DCA was used to evaluate the clinical effectiveness of the model in predicting EC-MI (use case 1).

Statistical analyses were conducted using two-tailed tests, and a $p$ value < 0.05 was considered statistically significant.

### Hardware and software

For deep learning tasks, the CPU(Intel(R) Xeon(R) Platinum 8358P CPU @ 2.60 GHz), the NVIDIA RTX A6000 graphics card with CUDA version 12.2 and 48GB of GPU memory was utilized. The deep learning framework employed was PyTorch 2.0.0+cu117, implemented in Python (version 3.9.18; http://www.python.org/). Additionally, MATLAB version 2021b was used for certain analysis tasks. Statistical tests were performed using SPSS (SPSS Statistics 26.0).

### Reporting summary

Further information on research design is available in the Nature Portfolio Reporting Summary linked to this article.

## Data availability

The EC dataset (use case1) in the current study are not publicly available for patient privacy policy. However, if researchers wish to access our data solely for scientific research purposes, access can be obtained by sending an email request to the corresponding author. Requests will be processed by the corresponding author within 3 months and followed up with the requesting party. Any requests will be pending prior approval and revision by the Ethics Committee of Jiangmen Central Hospital, the Ethics Committee of Yuebei People's Hospital, the Ethics Committee of Affiliated Dongguan Hospital Southern Medical University, the Ethics Committee of Maoming People's Hospital, the Ethics Committee of Kaiping Central Hospital and the Ethics Committee of the Third Affiliated Hospital of Guangzhou Medical University. The Breast Cancer Histology Image dataset (use case 2) used in this study are available in link: https://worksheets.codalab.org/rest/bundles/0xe45e15f39fb54e9d9e919556af67aabe/contents/blob/. The Prostate MRI dataset (use case 3) used in this study are available in link: https://liuquande.github.io/SAML/. The Histology Nuclei dataset (use case 4) used in this study are available in link: https://monusac-2020.grand-challenge.org/Data/; https://zenodo.org/record/1175282/files/TNBC_NucleiSegmentation.zip; https://monuseg.grand-challenge.org/Data/. The deidentified relevant data generated in this study are provided in the Supplementary Information/Source Data file and can be downloaded from the following link: https://pan.baidu.com/s/1ZOzXIsG3ez3F9xyxsKZD8g?pwd=cyww, with the access code: cyww. Source data are provided with this paper.

## Code availability

The codes are provided at GitHub (https://github.com/baofengguat/VFMGL/tree/main).

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

## Acknowledgements
This work was supported by the National Natural Science Foundation of China (82460361 to B.F., 62176104 to W.S.L, 12261027 to Q.G.J), Chongqing Big Data Collaborative Innovation Center(CQBDCIC202304 to B.F.). GUAT Special Research Projecton the Strategic Development of Distinctive Interdisciplinary Fields(TS2024231 to B.F.). The authors of this paper greatly appreciate the support and assistance provided by the Third Affiliated Hospital of Guangzhou Medical University in this study.

## Author contributions
S.L.L. conceived and designed the study, developed all methods, and drafted the manuscript. Y.H.C. implemented the methodology, conducted experiments, and contributed to manuscript writing. Y.C. participated in relevant medical research, and contributed to manuscript writing. P.J.L., J.Q.S., C.Y.Z., Y.J.Z., B.L., and M.W.L. collected multi-center experimental data and information. Q.G.J. provided assistance for engineering experiments. E.M.C. offered theoretical support for medical research. W.S.L. contributed to the design of the methodology and manuscript writing. B.F. provided overall support for the research and contributed to method design. All authors reviewed and revised the manuscript.

## Competing interests
The authors declare no competing interests.
