## [Transparent Peer Review file · Nature Communications]

General Lightweight Framework for Vision Foundation Model Supporting Multi-Task and Multi-Center Medical Image Analysis

Corresponding Author: Professor Bao Feng

Version 0:

Reviewer comments:

Reviewer #1

(Remarks to the Author)

The paper proposes a Vision Foundation Model General Lightweight (VFMGL) that is supposed to be an alternative to generally larger foundation models. VFMGL is supposed to help with "decentralized construction of expert clinical models for various medical task". The paper reports several use cases in histoathology and radiology to demonstrate the usefulness of VFMGL for 1) transferring general knowledge from large foundation models to lightweight models for specific medical tasks, (2) enhancing model performance by extracting generalizable features from multi-site data without data transfer, (3) providing interpretability of the model's decision-making.

Below several points in no particular order:

- Right at the start of Introduction the authors mention the necessity of "large amount of annotated medical data" for AI as a major issue. This is relative as self-supervision, as the backbone of foundation models do not depend on annotated data.
- Data heterogeneity is also mentioned as a major challenge. Although true, but this can be technologically solved by large foundation models (FMs) in a centralized manner.
- Only SAM and DINOv2 are mentioned as FMs. Recent FMs (such as Virchow in pathology and many more) have not been mentioned.
- Figure 1a is rather confusing, it seems the Figure wants to show that Centers A-D send their information through HGKT but the arrows show a different story.
- "The HGKT technique leverages general knowledge from open-source VFM to perform medical tasks, automatically matching model layers between open-source VFM and local models, facilitating knowledge transfer among heterogeneous models." The Student-Teacher scheme does that too. Is HGKT a Student-Teacher system?
- Although sensitivity and specificity are included in radar charts, adding F1-score would provide more nuanced information on precision and recall.
- The results have been benchmarked against other FL schemes (although major ones like ProxyFL are also missing). However, comparison with non-FL SoTA models (FM or not) are missing.

Reviewer #2

(Remarks to the Author)

1. What are the noteworthy results?

1.1 This study is designed to leverage multi-center data for two tasks (classification and segmentation). The key highlights of this research are as follows:

1.1.1 Lightweight model design: They designed a model for lightweight parameterization of foundation model parameters, achieving a compression of the model parameter size to one-eighth of its original size. Additionally, the theoretical computational workload is reduced to one-twelfth of its original value.

1.1.2 Cross-center generalization: The local diagnostic model improves cross-center generalization in the four utilization scenarios by integrating multi-center knowledge obtained through knowledge distillation from the shared model.

1.1.3 External validation set: They analyzed the similarity between the data distribution of independent centers and the VFMGL training set to select optimal local models for potential real-world applications.

2. Will the work be of significance to the field and related fields? How does it compare to the established literature? If the work is not original, please provide relevant references.

2.1 VFMGL framework presented in this paper stands out from the established literature in two key aspects. First, VFMGL tackles the challenge of data heterogeneity through its novel DDBL method for selecting low-heterogeneity data and enabling cross-center generalization. Second, while most research has focused on improving model performance, VFMGL achieves substantial computational efficiency through the HGKT transfer learning technique.

2.2 However, it is somewhat uncertain whether the generalization ability was obtained through DDBL's sample selection. To make this clear, ablation studies supporting the effectiveness of DDBL are required.

2.3 Furthermore, in the case of HGKT, the VFM uses a transformer-based model while the local model uses a CNN. However, there is a lack of explanation for why fundamentally different models were chosen and how a linear transformation can effectively address the structural and fundamental differences between these models. We have detailed our concerns in the following questions and hope the authors will address them.

3. Does the work support the conclusions and claims, or is additional evidence needed?

3.1 DDBL was proposed to select low-heterogeneity data, but what constitutes "low-heterogeneity data" and its rationale is unclear, it raises several important questions:

3.1.1 Doesn't this method reduce the model's ability to learn diverse features from various samples, thereby potentially limiting its generalization capability?

3.1.2 Is there a risk that the specific classes or samples might be excluded during this process?

3.1.3 It would be beneficial to see qualitative examples of data classified as low-heterogeneity and high-heterogeneity.

3.2 To verify these concerns and better understand the impact of DDBL, ablation studies are necessary to determine if the cross-center generalization ability was indeed achieved through Stage 2. These studies should include:

3.2.1 Evaluating generalization performance with Stage 1 alone.

3.2.2 Assessing generalization performance in Stage 2 without DDBL, focusing only on Knowledge Distillation (KD).

3.3 VFMGL's performance on robustness is impressive but could be improved by refining the experimental setup. The current division of the test set into two groups based on the age cutoff of 54.7 is not representative, as endometrial cancer typically affects women around 60. Using a more appropriate cutoff closer to this age would yield more relevant insights. Additionally, considering the higher incidence of endometrial cancer in Black women compared to White women, incorporating ethnicity into the analysis and conducting this experiment if possible would provide a more comprehensive understanding of demographic impacts.

3.4 Furthermore, since the authors used a private dataset for 'VFMGL exhibits robustness' experiments, conducting experiments using a public dataset would allow for future comparisons with other studies. If demographic information is not available for the public dataset, which makes part 1 of the experiment infeasible, conducting part 2 of the experiment would still be valuable.

4. Are there any flaws in the data analysis, interpretation and conclusions? Do these prohibit publication or require revision?

4.1 N/A

5. Is the methodology sound? Does the work meet the expected standards in your field?

5.1 In the DDBL process, outlier samples are excluded to maintain low heterogeneity. Given that only features corresponding to a similar data distribution are selected, what is the rationale for claiming this process contributes to generalizability?

5.2 Why does VFM use a transformer-based model while the local model uses a CNN? Despite the differences in the learning mechanisms between the two models, there seems to be some mismatch. An explanation is needed on why the weight transformer could still be effective in this context.

5.3 The authors emphasize the computational benefits of the lightweight local models in VFMGL. However, the overall process requires a complex training procedure, resulting in substantial computational costs. Notably, the final outcome is a

separate local model for each medical center and task, resulting in a total model count of (# of centers) x (# of tasks). Although the individual models are lightweight, the cumulative cost of constructing models is still high. The authors should address this limitation and discuss strategies to reduce the overall model construction overhead or examine the tradeoffs between individual model efficiency and total modeling costs.

6. Is there enough detail provided in the methods for the work to be reproduced?

6.1 Page 5, lines 122 to 123: How was the similarity of data distribution between data centers measured?

6.2 What do the values on the x and y axes of the heatmap in Fig.10 represent? If the alphabet represents 'data center,' do the numbers following it refer to samples or to one of the features among the 256 features?

6.3 In the supplementary materials, what do θ_{wfm} , L_{wfm} , and L_{org} mean? Does wfm and org refer to the local model?

6.4 The paper lacks training details (e.g., which models were used for open-source VFM, shared model, and local models, batch size, iteration steps, and other hyper-parameters).

6.4.1 Specifically, regarding the weight transfer strategy, what VFM model was used? The supplementary table S7 notes that DINO-V2 was used as the source model and ResNet-18 as the expert model for clinical diagnosis in use case 1. Is this setting fixed?

6.4.2 When performing classification and segmentation, is the same VFM model used for both tasks?

6.4.3 For segmentation, on which model is it based? If ResNet-18 is used as the encoder, what model is used as the decoder?

Reviewer #3

(Remarks to the Author)

This paper addresses the efficiency of deep foundational models for medical image diagnosis. It primarily focuses on distilling open-sourced vision transformers (ViTs) and transferring the distilled knowledge to local convolutional neural networks (CNNs). Additionally, this work explores topics such as federated learning to ensure model training on patient-sensitive data. However, in my opinion, mechanically combining both model distillation and federated learning is neither novel nor particularly useful.

- The concept of distilling CNNs from ViTs has been extensively researched since 2022 [Yao, Xufeng, et al. "Distill Vision Transformers to CNNs via Low-Rank Representation Approximation."]. Furthermore, similar and more intuitive research on distilling a CNN to train a ViT better has been thoroughly investigated by the famous DeiT paper. These previous works significantly challenge the novelty of this paper.

- This paper introduces its distillation losses. What is the theoretical basis for these losses? How does their performance compare with state-of-the-art methods, especially the work by Hinton et al. [Hinton, Geoffrey, Oriol Vinyals, and Jeff Dean. "Distilling the knowledge in a neural network." arXiv preprint arXiv:1503.02531 (2015)]?

- The ROC-AUC curve in Figure 2 indicates that there are not enough samples in the test set, which raises questions about the reliability of the authors' results.

- The motivation for this study might not be valid. The size of modern transformers is not as cumbersome as suggested by the authors. For example, Swin-T models have roughly the same size of parameters as ResNet-50.

Version 1:

Reviewer comments:

Reviewer #3

(Remarks to the Author)

I think the authors had carefully replied to all my comments previously. I have nothing to add.

Reviewer #4

(Remarks to the Author)

The authors have provided a comprehensive response to the reviewers' comments, addressing all points effectively.

Reviewer #5

(Remarks to the Author)

What are the noteworthy results?

The authors propose a light-weight framework adapted to Vision Foundation Models in a medical image analysis setup.

Will the work be of significance to the field and related fields? How does it compare to the established literature? If the work is not original, please provide relevant references.

No. The proposed method makes incremental contributions by addressing engineering challenges when using Vision Foundation Models in multi-center setups. The scientific problem behind is not that significant. Additionally, the proposed method shall also work in non-medical scenarios.

Does the work support the conclusions and claims, or is additional evidence needed?

Yes.

Are there any flaws in the data analysis, interpretation and conclusions? Do these prohibit publication or require revision?

Not found. The authors have extensively revised the manuscript in terms of additional results.

Is the methodology sound? Does the work meet the expected standards in your field?

The work does show merit for a computer science or engineering journal. But for Nature Communications, the scientific and practical clinical value requires further justifications. Based on my understanding, the proposed method is evaluated based on simulation. No real system is deployed yet.

Is there enough detail provided in the methods for the work to be reproduced?

I believe so, but I didn't check the details.

Reviewer #1 (Remarks to the Author):

The paper proposes a Vision Foundation Model General Lightweight (VFMGL) that is supposed to be an alternative to generally larger foundation models. VFMGL is supposed to help with "decentralized construction of expert clinical models for various medical task". The paper reports several use cases in histoathology and radiology to demonstrate the usefulness of VFMGL for 1) transferring general knowledge from large foundation models to lightweight models for specific medical tasks, (2) enhancing model performance by extracting generalizable features from multi-site data without data transfer, (3) providing interpretability of the model's decision-making.

Response:

Thank you very much for your attention and constructive feedback on our manuscript.

Below several points in no particular order:

- Right at the start of Introduction the authors mention the necessity of "large amount of annotated medical data" for AI as a major issue. This is relative as self-supervision, as the backbone of foundation models do not depend on annotated data.

Response:

Thank you for the valuable comments from the reviewer.

We agree with your perspective. We also recognize that certain statements in the introduction may have caused some confusion. Based on your feedback, we have refined the relevant expressions in the introduction and provided further clarification.

Self-supervised learning allows models to leverage large amounts of unlabeled data for pretraining, thereby reducing dependence on labeled data and significantly lowering data annotation costs. Foundation models typically employ self-supervised learning algorithms on large-scale unlabeled datasets to learn latent, generalized representations of data, making them suitable for various downstream tasks.

However, in the medical domain, there are significant differences between natural and medical images. As such, directly applying vision foundation models like SAM and DINOv2 to medical tasks may not meet performance requirements [1,2,3]. Additionally, when applying vision foundation models to specific downstream tasks, a large amount of annotated data is still needed for fine-tuning or constructing downstream classifiers [1,2]. For example, Jun Ma et al. [1] used over a million pairs of medical image-mask data to fine-tune SAM for medical segmentation tasks, demonstrating a significant performance improvement in the SAM model after fine-tuning with labeled data. The Virchow model [2] pretrained on 1.5 million whole-slide histopathological images used model embeddings and around 80,000 annotated medical data instances to construct a classifier, achieving excellent performance in specific downstream tasks. Furthermore, in tile-level performance validation, the study used nearly one million tiles meticulously annotated by pathologists to train the Virchow model, achieving high performance in tile-level predictions.

Therefore, this study proposes a VFMGL framework suitable for multi-task, multi-center applications, enabling automated customization of model layer matching and feature transfer. This framework allows local models at each center to acquire general knowledge from open-source VFMs, facilitating robust feature extraction without data sharing. By leveraging redundant parameters in local models to learn common knowledge across centers, this approach further enhances model generalization performance. The framework's efficacy is validated across various

tasks and datasets, with demonstrable interpretability.

- Data heterogeneity is also mentioned as a major challenge. Although true, but this can be technologically solved by large foundation models (FMs) in a centralized manner.

Response:

Data heterogeneity can lead to reduced model performance across different datasets. Therefore, when dealing with heterogeneous medical data, attention must be given to the robustness and generalization performance of the model. In a centralized server or computing environment, training large foundation models using substantial medical data enables the model to learn generalizable features suited to the medical field, thereby mitigating the impact of data heterogeneity on model performance.

It is important to note that, unlike natural data, medical data is typically subject to stringent privacy protections and legal regulations, making large-scale, centralized data sharing challenging [4,5]. Thus, while foundation models in other fields may address heterogeneity through centralized environments, in the medical domain, the centralized collection and training of large-scale medical data still face both technical and legal obstacles. Additionally, although foundation models are capable of handling heterogeneous data, specific medical tasks often demand highly specialized requirements. Therefore, foundation models still require fine-tuning and embedding to accommodate various medical application scenarios [1,2,3]. In response to these challenges, the aim of this study is to construct a model with robust generalization capabilities in a decentralized manner.

- Only SAM and DINOv2 are mentioned as FMs. Recent FMs (such as Virchow in pathology and many more) have not been mentioned.

Response:

We appreciate your feedback, which has helped us further refine and enrich our paper.

Following your suggestions and based on our understanding of representative studies on medical vision foundation models (VFMs), we have supplemented the third paragraph of the introduction with descriptions of research on medical VFMs (e.g., Virchow, MedSAM).

Jun Ma et al.[1] fine-tuned MedSAM using over 20 A100 GPUs with a total memory capacity of 80GB, employing an annotated dataset of over one million medical images. As a prompt-based, semi-automated segmentation method, MedSAM demonstrated excellent performance across multiple medical image segmentation tasks. Eugene Vorontsov et al.[2] pretrained the Virchow model using 1.5 million whole-slide histopathology images and subsequently constructed a classifier based on model embeddings and nearly 80,000 annotated medical cases, achieving outstanding performance on specific downstream tasks. Training, fine-tuning, and deploying VFMs typically require extensive data support and consume substantial hardware resources and time.

- Figure 1a is rather confusing, it seems the Figure wants to show that Centers A-D send their information through HGKT but the arrows show a different story.

Response:

We thank you for pointing out this important issue.

Based on your suggestions, we have further refined the visual and textual elements of Figure

1a and revised the content under the title "Vision Foundation Model General Lightweight Framework" to enhance readability. In Figure 1a, each center retains its own private medical data and, using the HGKT method, transfers general knowledge from open-source VFMs to develop a lightweight, robust local model. Subsequently, following the DDBL strategy, we select low-heterogeneity data from each center. Through knowledge distillation, we enable the redundant parameters of the local models to learn common knowledge from the shared model, thereby further enhancing the cross-center generalization capability of the lightweight local models.

Fig.1 Overview of the VFMGL framework. (a)Construction of Local and shared Models. (b) Construction of robustness critical layers based on HGKT and general knowledge from VFM. (c) Continued model construction based on DDBL and common knowledge. Notes: VFMGL, Vision Foundation Model General Lightweight; HGKT, Heterogeneous model General Knowledge Transfer; DDBL, Data Deduction in Batch-Level; VFM, Vision Foundation Model.

- "The HGKT technique leverages general knowledge from open-source VFM to perform medical tasks, automatically matching model layers between open-source VFM and local models, facilitating knowledge transfer among heterogeneous models." The Student-Teacher scheme does that too. Is HGKT a Student-Teacher system?

Response:

Thank you for the valuable comments from the reviewer.

HGKT is a type of Student-Teacher system. The Student-Teacher system transfers knowledge from a teacher model, with a large number of parameters, to a student model, thereby reducing training costs, enhancing student model performance, and minimizing parameter counts. The student model inherits the teacher model's capability to express visual features by learning either the intermediate layer representations or the output probabilities from the teacher model on the same data.

VFMs possess a large number of parameters and are trained on massive natural image datasets using self-supervised methods, endowing them with strong robust and generalized representation capabilities [6]. These robust visual features of VFMs help reduce the impact of data heterogeneity on model performance. In this study, given the substantial domain differences between natural and medical images, blindly matching model layers or transferring knowledge may not benefit medical tasks and may even result in negative transfer. Some researchers select specific intermediate layers of models based on experience or experiments to conduct layer matching and knowledge transfer [4,7,8]. This process is both time-consuming and labor-intensive, especially for VFMs with their extensive model layers. With changes in tasks and datasets, the validity of model layers selected through manual methods remains uncertain. Additionally, in multi-center scenarios, model knowledge adapted to data in one center may differ due to data heterogeneity [5], and correspondingly, the knowledge required from VFMs by each center may also vary.

Therefore, we propose the HGKT method. In helping each center transfer generalized knowledge from open-source VFMs to local models (as shown in Fig.1b), we employ a meta-network to adaptively compute layer matching weights and feature transfer weights. Each center has its own customized weights, enabling tailored layer matching and feature transfer. Table S15 displays the matching weights for each model layer, indicating substantial differences in the degree of layer matching between local models and VFMs across centers, supporting the aforementioned perspective. HGKT achieves automated layer matching and feature transfer between heterogeneous models, saving considerable manual effort in multi-task, multi-center scenarios. Using the HGKT method, local models transfer generalized knowledge from VFMs, gaining robust feature extraction capabilities.

- Although sensitivity and specificity are included in radar charts, adding F1-score would provide more nuanced information on precision and recall.

Response:

Thanks for your thoughtful suggestion.

Based on your suggestions, we have added F1-scores in use cases 1–4 to enhance the completeness of the results (Fig. 2, Fig. 3, Fig. 4, Fig. 6). In Case 1, the F1 scores of VFMGL at the various centers were 0.905, 0.703, 0.889, and 0.849, respectively. In Case 2, the F1 scores of VFMGL at the centers were 0.9889, 0.9721, 0.9913, 0.9699, and 0.9883, respectively. In Case 3, the F1 scores of VFMGL at the centers were 0.8831, 0.8729, 0.9302, 0.8402, 0.8881, and 0.8447, respectively. In Case 4, the F1 scores of VFMGL at the centers were 0.6796, 0.6706, 0.6852, 0.6326, 0.6596, and 0.6732, respectively. The results indicate that VFMGL demonstrates good F1 performance in classification and segmentation across various datasets. More detailed results are available in Tables S1–S4.

- The results have been benchmarked against other FL schemes (although major ones like ProxyFL are also missing). However, comparison with non-FL SoTA models (FM or not) are missing.

Response:

Thank you for your feedback, which has helped us further refine the results of our paper.

Following your suggestions, we have continued to enhance our work by adding two foundation models (FMs), Virchow (for classification) and MedSAM (for segmentation), as comparison methods. ProxyFL is an outstanding framework addressing efficient communication and privacy protection, allowing effective information exchange among participants by constructing proxy models and utilizing differential privacy analysis for proxy communication, thereby achieving enhanced privacy guarantees. We have supplemented the Methods section with this work to improve the description of FL-related approaches.

For the classification tasks, we follow the usage recommended in the Virchow paper, training models based on Virchow embeddings and annotated training data to adapt them to specific classification tasks. In the endometrial cancer myometrial invasion classification task (use case 1), VFMGL demonstrated an average AUC improvement across centers of approximately 8.9%, 6.6%, 11.4%, and 5.8% compared to other methods (Fig. 2). Additionally, compared to Virchow, VFMGL's overall AUC improved by around 5.2% (for more details, see Table S1). In the breast cancer histology image classification task (use case 2), VFMGL achieved an average AUC improvement across centers of approximately 2.3%, 5.0%, 2.0%, 1.4%, and 1.5%, with an F1 score improvement of approximately 4.8%, 8.2%, 5.0%, 4.7%, and 3.5% (Fig. 3). Compared to Virchow, VFMGL's overall AUC improved by about 2.7% (see Table S2 for more details). Compared with the foundation model (FM), VFMGL still exhibited excellent performance in multi-center classification tasks.

For segmentation tasks, MedSAM, a prompt-based semi-automatic segmentation model, was used. We accurately calculated the position and size of the prompt box using real mask labels (with the segmentation target at the center of the prompt box) and applied slight random resizing to simulate usage scenarios (a 10-pixel random resize for use case 3 and a 5-pixel random resize for use case 4). In the prostate segmentation task (use case 3), VFMGL achieved an average improvement of approximately 2.8% in dice performance and 3.5% in F1 score across centers compared to MedSAM (Fig. 4). Further details can be found in Table S3. In the nuclear segmentation task (use case 4), as each image contained numerous segmentation targets, the prompt box position was calculated via masks. MedSAM exhibited excellent performance across centers A-D and F. Results indicated that VFMGL's fully automatic segmentation performance was comparable to the semi-automatic segmentation (Fig. 6), and VFMGL demonstrated strong stability when handling data heterogeneity in multi-center scenarios. More detailed results are available in Table S4.

Fig.2 ROC curves, DCA curves and radar charts for the four centers. (A) ROC curves of five models in four centers. (B) DCA curves of five models in four centers. (C) Radar chart comparison of five models in four centers. Notes: VFMGL, Vision Foundation Model General Lightweight; AUC, area under the curve; TPR, True Positive Rate; FPR, False Positive Rate; PPV, positive predictive value; NPV, negative predictive value.

Fig.3 Radar chart comparison of five models in five centers. Notes: VFMGL, Vision Foundation Model General Lightweight. AUC, area under the curve; PPV, positive predictive value; NPV, negative predictive value.

Fig.4 Radar charts for the six centers. Notes: VFMGL, Vision Foundation Model General Lightweight. IOU, intersection over union; PPV, positive predictive value.

Fig.6 Radar charts for the six centers. Notes: VFMGL, Vision Foundation Model General Lightweight. IOU, intersection over union; PPV, positive predictive value.

Table S1. use case 1

Center	Methods	AUC	Sensitivity	Specificity	Accuracy	PPV	NPV	F1
A	FedAvg	0.676	0.461(111/241)	0.794(27/34)	0.502(138/275)	0.941(111/118)	0.172(27/157)	0.619

	FedProx	0.702	0.448(108/241)	0.824(28/34)	0.495(136/275)	0.947(108/114)	0.174(28/161)	0.608
	HarmoFL	0.719	0.685(165/241)	0.647(22/34)	0.680(187/275)	0.932(165/177)	0.224(22/98)	0.790
	MetaFed	0.732	0.705(170/241)	0.647(22/34)	0.698(192/275)	0.934(170/182)	0.237(22/93)	0.804
	Virchow	0.716	0.622(150/241)	0.706(24/34)	0.633(174/275)	0.938(150/160)	0.209(24/115)	0.748
	VFMGL	0.798	0.884(213/241)	0.500(17/34)	0.836(230/275)	0.926(213/230)	0.378(17/45)	0.905
B	FedAvg	0.710	0.696(16/23)	0.500(3/6)	0.655(19/29)	0.842(16/19)	0.300(3/10)	0.762
	FedProx	0.754	0.739(17/23)	0.667(4/6)	0.724(21/29)	0.895(17/19)	0.400(4/10)	0.810
	HarmoFL	0.790	0.739(17/23)	0.667(4/6)	0.724(21/29)	0.895(17/19)	0.400(4/10)	0.810
	MetaFed	0.775	0.609(14/23)	0.667(4/6)	0.621(18/29)	0.875(14/16)	0.308(4/13)	0.718
	Virchow	0.804	0.826(19/23)	0.667(4/6)	0.793(23/29)	0.905(19/21)	0.500(4/8)	0.864
	VFMGL	0.833	0.565(13/23)	0.833(5/6)	0.621(18/29)	0.929(13/14)	0.333(5/15)	0.703
C	FedAvg	0.643	0.571(8/14)	1.000(1/1)	0.600(9/15)	1.000(8/8)	0.143(1/7)	0.727
	FedProx	0.714	0.714(10/14)	0.000(0/1)	0.667(10/15)	0.909(10/11)	0.000(0/4)	0.800
	HarmoFL	0.786	0.929(13/14)	0.000(0/1)	0.867(13/15)	0.929(13/14)	0.000(0/1)	0.929
	MetaFed	0.786	0.786(11/14)	1.000(1/1)	0.800(12/15)	1.000(11/11)	0.250(1/4)	0.880
	Virchow	0.786	0.857(12/14)	0.000(0/1)	0.800(12/15)	0.923(12/13)	0.000(0/2)	0.889
	VFMGL	0.857	0.857(12/14)	0.000(0/1)	0.800(12/15)	0.923(12/13)	0.000(0/2)	0.889
D	FedAvg	0.765	0.882(97/110)	0.500(4/8)	0.856(101/118)	0.960(97/101)	0.235(4/17)	0.919
	FedProx	0.752	0.500(55/110)	1.000(8/8)	0.534(63/118)	1.000(55/55)	0.127(8/63)	0.667
	HarmoFL	0.805	0.582(64/110)	0.875(7/8)	0.602(71/118)	0.985(64/65)	0.132(7/53)	0.732
	MetaFed	0.807	0.836(92/110)	0.500(4/8)	0.814(96/118)	0.958(92/96)	0.182(4/22)	0.893
	Virchow	0.823	0.636(70/110)	0.875(7/8)	0.653(77/118)	0.986(70/71)	0.149(7/47)	0.773
	VFMGL	0.848	0.745(82/110)	0.875(7/8)	0.754(89/118)	0.988(82/83)	0.200(7/35)	0.849

Notes: VFMGL, vision foundation model general lightweight; AUC, area under the curve; PPV, positive predictive value; NPV, negative predictive value.

Table S2. use case 2

Center	Methods	AUC	Sensitivity	Specificity	Accuracy	PPV	NPV	F1
A	FedAvg	0.9358	0.8484 (5043/5944)	0.9561 (5683/5944)	0.9023 (10726/11888)	0.9508 (5043/5304)	0.8632 (5683/6584)	0.8967
	FedProx	0.9875	0.9263 (5506/5944)	0.9739 (5789/5944)	0.9501 (11295/11888)	0.9726 (5506/5661)	0.9297 (5789/6227)	0.9489
	HarmoFL	0.9863	0.9127 (5425/5944)	0.9754 (5798/5944)	0.9441 (11223/11888)	0.9738 (5425/5571)	0.9178 (5798/6317)	0.9423
	MetaFed	0.9874	0.9268 (5509/5944)	0.9798 (5824/5944)	0.9533 (11333/11888)	0.9787 (5509/5629)	0.9305 (5824/6259)	0.9520
	Virchow	0.9831	0.9633 (5726/5944)	0.9680 (5754/5944)	0.9657 (11480/11888)	0.9679 (5726/5916)	0.9635 (5754/5972)	0.9656
	VFMGL	0.9992	0.9882 (5874/5944)	0.9896 (5882/5944)	0.9889 (11756/11888)	0.9896 (5874/5936)	0.9882 (5882/5952)	0.9889
	FedAvg	0.8420	0.7946 (2773/3490)	0.8479 (2960/3491)	0.8212 (5733/6981)	0.8393 (2773/3304)	0.8050 (2960/3677)	0.8163
B	FedProx	0.9711	0.9060 (3162/3490)	0.9215 (3217/3491)	0.9138 (6379/6981)	0.9203 (3162/3436)	0.9075 (3217/3545)	0.9131
	HarmoFL	0.9658	0.8837 (3084/3490)	0.9184 (3206/3491)	0.9010 (6290/6981)	0.9154 (3084/3369)	0.8876 (3206/3612)	0.8993
	MetaFed	0.9776	0.9095 (3174/3490)	0.9433 (3293/3491)	0.9264 (6467/6981)	0.9413 (3174/3372)	0.9124 (3293/3609)	0.9251
	Virchow	0.9795	0.8123 (2835/3490)	0.9989 (3487/3491)	0.9056 (6322/6981)	0.9986 (2835/2839)	0.8419 (3487/4142)	0.8959
	VFMGL	0.9973	0.9481 (3309/3490)	0.9974 (3482/3491)	0.9728 (6791/6981)	0.9973 (3309/3318)	0.9506 (3482/3663)	0.9721

C	FedAvg	0.9713	0.9155 (7787/8506)	0.9509 (8087/8505)	0.9332 (15874/17011)	0.9491 (7787/8205)	0.9184 (8087/8806)	0.9320
	FedProx	0.9892	0.9474 (8059/8506)	0.9714 (8262/8505)	0.9594 (16321/17011)	0.9707 (8059/8302)	0.9487 (8262/8709)	0.9589
	HarmoFL	0.9829	0.9319 (7927/8506)	0.9566 (8136/8505)	0.9443 (16063/17011)	0.9555 (7927/8296)	0.9336 (8136/8715)	0.9436
	MetaFed	0.9886	0.9452 (8040/8506)	0.9688 (8240/8505)	0.9570 (16280/17011)	0.9681 (8040/8305)	0.9465 (8240/8706)	0.9565
	Virchow	0.9670	0.8594 (7310/8506)	0.9805 (8339/8505)	0.9199 (15649/17011)	0.9778 (7310/7476)	0.8746 (8339/9535)	0.9148
	VFMGL	0.9995	0.9911 (8430/8506)	0.9915 (8433/8505)	0.9913 (16863/17011)	0.9915 (8430/8502)	0.9911 (8433/8509)	0.9913
D	FedAvg	0.9763	0.9067 (11772/12984)	0.9546 (12394/12984)	0.9306 (24166/25968)	0.9523 (11772/12362)	0.9109 (12394/13606)	0.9289
	FedProx	0.9925	0.9478 (12306/12984)	0.9784 (12704/12984)	0.9631 (25010/25968)	0.9778 (12306/12586)	0.9493 (12704/13382)	0.9626
	HarmoFL	0.9856	0.9216 (11966/12984)	0.9718 (12618/12984)	0.9467 (24584/25968)	0.9703 (11966/12332)	0.9253 (12618/13636)	0.9453
	MetaFed	0.9927	0.9477 (12305/12984)	0.9823 (12754/12984)	0.9650 (25059/25968)	0.9817 (12305/12535)	0.9495 (12754/13433)	0.9644
	Virchow	0.9736	0.6842 (8883/12984)	0.9992 (12973/12984)	0.8417 (21856/25968)	0.9988 (8883/8894)	0.7598 (12973/17074)	0.8120
	VFMGL	0.9977	0.9427 (12240/12984)	0.9988 (12969/12984)	0.9708 (25209/25968)	0.9988 (12240/12255)	0.9457 (12969/13713)	0.9699
E	FedAvg	0.9827	0.9439 (13849/14672)	0.9567 (14038/14673)	0.9503 (27887/29345)	0.9562 (13849/14484)	0.9446 (14038/14861)	0.9500
	FedProx	0.9946	0.9600 (14085/14672)	0.9808 (14392/14673)	0.9704 (28477/29345)	0.9804 (14085/14366)	0.9608 (14392/14979)	0.9701
	HarmoFL	0.9912	0.9624 (14121/14672)	0.9554 (14019/14673)	0.9589 (28140/29345)	0.9557 (14121/14775)	0.9622 (14019/14570)	0.9590
	MetaFed	0.9947	0.9625 (14122/14672)	0.9761 (14322/14673)	0.9693 (28444/29345)	0.9757 (14122/14473)	0.9630 (14322/14872)	0.9691
	Virchow	0.9589	0.8676 (12729/14672)	0.9826 (14417/14673)	0.9251 (27146/29345)	0.9803 (12729/12985)	0.8812 (14417/16360)	0.9205
	VFMGL	0.9993	0.9845 (14445/14672)	0.9922 (14559/14673)	0.9884 (29004/29345)	0.9922 (14445/14559)	0.9846 (14559/14786)	0.9883

Notes: VFMGL, vision foundation model general lightweight; AUC, area under the curve; PPV, positive predictive value; NPV, negative predictive value.

Table S3. use case 3

Center	Methods	Dice	ASSD	Sensitivity	Specificity	PPV	IOU	F1
A	FedAvg	0.5183	55.6626	0.7629	0.7337	0.1240	0.1186	0.2133
	FedProx	0.8268	23.4177	0.6776	0.9894	0.7460	0.5468	0.7102
	HarmoFL	0.9253	6.7873	0.8624	0.9951	0.8711	0.7481	0.8667
	MetaFed	0.7531	23.1151	0.5629	0.9897	0.7364	0.4758	0.6381
	MedSAM	0.9163	5.3587	0.7665	0.9989	0.9682	0.7399	0.8556
	VFMGL	0.9340	5.6570	0.8981	0.9953	0.8685	0.7872	0.8831
B	FedAvg	0.5034	52.5426	0.8354	0.7513	0.0840	0.0827	0.1527
	FedProx	0.8879	15.7715	0.8247	0.9929	0.7207	0.6001	0.7692
	HarmoFL	0.9390	7.5250	0.9570	0.9935	0.7534	0.7162	0.8431
	MetaFed	0.8995	8.4520	0.8583	0.9948	0.7854	0.6682	0.8202
	MedSAM	0.8810	4.8425	0.6994	0.9993	0.9614	0.6651	0.8098
	VFMGL	0.9328	5.0626	0.9337	0.9956	0.8196	0.7659	0.8729
C	FedAvg	0.7868	25.7947	0.9805	0.9543	0.4144	0.4060	0.5826
	FedProx	0.9037	6.6428	0.8132	0.9968	0.8836	0.7172	0.8469
	HarmoFL	0.9525	4.5665	0.9597	0.9939	0.8232	0.7895	0.8862
	MetaFed	0.8913	9.2436	0.8297	0.9912	0.7496	0.6494	0.7876
	MedSAM	0.8965	5.7118	0.7270	0.9988	0.9594	0.6929	0.8272

D	VFMGL	0.9620	2.8192	0.9539	0.9973	0.9076	0.8672	0.9302
	FedAvg	0.6957	45.0055	0.7222	0.9320	0.2818	0.2584	0.4054
	FedProx	0.8536	18.7689	0.7450	0.9919	0.7242	0.5928	0.7345
	HarmoFL	0.9084	13.8240	0.9455	0.9892	0.7334	0.6855	0.8261
	MetaFed	0.8716	15.8724	0.7776	0.9880	0.6874	0.5869	0.7297
	MedSAM	0.9098	4.2473	0.7565	0.9995	0.9840	0.7416	0.8554
E	VFMGL	0.9191	9.4369	0.8434	0.9966	0.8370	0.7465	0.8402
	FedAvg	0.7557	44.8552	0.8460	0.9364	0.3739	0.3555	0.5186
	FedProx	0.9018	14.1736	0.8811	0.9896	0.7645	0.6925	0.8187
	HarmoFL	0.9345	6.8490	0.8794	0.9948	0.8569	0.7654	0.8680
	MetaFed	0.8151	19.1216	0.6407	0.9884	0.7034	0.5449	0.6706
	MedSAM	0.9129	4.9324	0.7698	0.9985	0.9661	0.7403	0.8569
F	VFMGL	0.9383	6.0202	0.9296	0.9946	0.8501	0.8028	0.8881
	FedAvg	0.5078	53.7082	0.7025	0.7794	0.0880	0.0856	0.1564
	FedProx	0.7754	27.9849	0.7216	0.9880	0.6303	0.5371	0.6729
	HarmoFL	0.9167	12.9202	0.8876	0.9897	0.7471	0.7012	0.8113
	MetaFed	0.8268	18.1592	0.6701	0.9874	0.6345	0.5282	0.6518
	MedSAM	0.9076	4.3989	0.7476	0.9994	0.9644	0.7154	0.8423
	VFMGL	0.9080	6.8139	0.8981	0.9942	0.7973	0.7445	0.8447

Notes: VFMGL, vision foundation model general lightweight; ASSD, average symmetric surface distance; PPV, positive predictive value; IOU: intersection over union.

Table S4. use case 4

Center	Methods	Dice	ASSD	Sensitivity	Specificity	PPV	IOU	F1
A	FedAvg	0.4493	5.5991	0.3041	0.5673	0.2572	0.1620	0.2787
	FedProx	0.7175	4.0955	0.6210	0.9322	0.7564	0.4858	0.6820
	HarmoFL	0.7126	4.1270	0.5655	0.9516	0.7929	0.4597	0.6602
	MetaFed	0.7066	4.2787	0.5823	0.9420	0.7715	0.4641	0.6637
	MedSAM	0.7703	1.8837	0.6338	0.8752	0.7653	0.5256	0.6934
	VFMGL	0.7509	3.6429	0.5774	0.9606	0.8257	0.4918	0.6796
B	FedAvg	0.4740	5.3538	0.3170	0.6262	0.2570	0.1647	0.2839
	FedProx	0.7140	4.3436	0.6542	0.9075	0.6687	0.4589	0.6614
	HarmoFL	0.7165	4.3308	0.5974	0.9358	0.7194	0.4509	0.6527
	MetaFed	0.6894	4.4940	0.6227	0.9056	0.6627	0.4413	0.6421
	MedSAM	0.7666	2.4434	0.6379	0.8743	0.6937	0.5073	0.6646
	VFMGL	0.7658	4.1422	0.6245	0.9347	0.7240	0.4731	0.6706
C	FedAvg	0.4336	10.2695	0.2468	0.5464	0.1723	0.1117	0.2029
	FedProx	0.7643	8.6537	0.7056	0.9073	0.6443	0.4772	0.6736
	HarmoFL	0.7696	8.4898	0.6516	0.9345	0.6886	0.4711	0.6696
	MetaFed	0.7199	8.7500	0.6851	0.8927	0.6213	0.4703	0.6516
	MedSAM	0.8219	2.5373	0.6901	0.9297	0.7839	0.5936	0.7340
	VFMGL	0.7735	8.2582	0.6203	0.9646	0.7653	0.4823	0.6852
D	FedAvg	0.4663	5.8823	0.3054	0.5668	0.2514	0.1587	0.2758

	FedProx	0.7080	4.3680	0.5639	0.9435	0.7748	0.4704	0.6527
	HarmoFL	0.6983	4.2005	0.5113	0.9616	0.8142	0.4459	0.6281
	MetaFed	0.6723	4.5729	0.5402	0.9410	0.7670	0.4466	0.6339
	MedSAM	0.7911	2.1497	0.6590	0.9024	0.7635	0.5534	0.7074
	VFMGL	0.7410	3.6780	0.5059	0.9681	0.8440	0.4471	0.6326
E	FedAvg	0.5288	1.0811	0.3998	0.6715	0.3341	0.2216	0.3640
	FedProx	0.6645	1.2186	0.5603	0.8924	0.6787	0.4406	0.6138
	HarmoFL	0.6901	1.1861	0.5380	0.9349	0.7690	0.4616	0.6331
	MetaFed	0.7016	1.1509	0.5698	0.9178	0.7382	0.4713	0.6432
	MedSAM	0.6591	1.2267	0.5590	0.7690	0.5127	0.3648	0.5349
	VFMGL	0.7568	1.0707	0.6386	0.8806	0.6821	0.4914	0.6596
F	FedAvg	0.4406	10.1532	0.3125	0.5589	0.1315	0.1010	0.1851
	FedProx	0.6677	7.3864	0.6236	0.8654	0.4064	0.2877	0.4921
	HarmoFL	0.6961	6.8815	0.6171	0.9039	0.4793	0.3260	0.5395
	MetaFed	0.6834	6.9784	0.6559	0.8965	0.4763	0.3391	0.5519
	MedSAM	0.8157	1.2948	0.6114	0.9693	0.7693	0.5180	0.6813
	VFMGL	0.7899	4.1967	0.6218	0.9729	0.7339	0.4772	0.6732

Notes: VFMGL, vision foundation model general lightweight; ASSD, average symmetric surface distance; PPV, positive predictive value; IOU: intersection over union.

References

- [1] Ma, J., He, Y., Li, F. et al. Segment anything in medical images. *Nat Commun* 15, 654 (2024). <https://doi.org/10.1038/s41467-024-44824-z>
- [2] Vorontsov, E., Bozkurt, A., Casson, A. et al. A foundation model for clinical-grade computational pathology and rare cancers detection. *Nat Med* 30, 2924–2935 (2024). <https://doi.org/10.1038/s41591-024-03141-0>
- [3] Xu, H., Usuyama, N., Bagga, J. et al. A whole-slide foundation model for digital pathology from real-world data. *Nature* 630, 181–188 (2024). <https://doi.org/10.1038/s41586-024-07441-w>
- [4] Chen YQ, Lu W, Qin X, Wang JD and Xie X. MetaFed: Federated Learning among Federations with Cyclic Knowledge Distillation for Personalized Healthcare. *IJCAI'22 federated learning workshop*.
- [5] Jiang MR, Wang ZR, Dou Q. HarmoFL: Harmonizing Local and Global Drifts in Federated Learning on Heterogeneous Medical Images. *Proceedings of the 36th AAAI Conference on Artificial Intelligence*. **36**, 914-922 (2022).
- [6] Maxime Oquab, Timothée Darcet, Théo Moutakanni. et al. DINOv2: Learning Robust Visual Features without Supervision. Preprint at <https://arxiv.org/pdf/2304.07193> (2023)
- [7] Romero, A., Ballas, N., Kahou, S. E., Chassang, A., Gatta, C., & Bengio, Y. Fitnets: Hints for thin deep nets. *ICLR* 2015.
- [8] Wang X , Fu T , Liao S ,et al. Exclusivity-Consistency Regularized Knowledge Distillation for Face Recognition[C]//European Conference on Computer Vision. Springer, Cham, 2020.DOI:10.1007/978-3-030-58586-0_20.

Reviewer #2 (Remarks to the Author):

1. What are the noteworthy results?

1.1 This study is designed to leverage multi-center data for two tasks (classification and segmentation). The key highlights of this research are as follows:

1.1.1 Lightweight model design: They designed a model for lightweight parameterization of foundation model parameters, achieving a compression of the model parameter size to one-eighth of its original size. Additionally, the theoretical computational workload is reduced to one-twelfth of its original value.

1.1.2 Cross-center generalization: The local diagnostic model improves cross-center generalization in the four utilization scenarios by integrating multi-center knowledge obtained through knowledge distillation from the shared model.

1.1.3 External validation set: They analyzed the similarity between the data distribution of independent centers and the VFMGL training set to select optimal local models for potential real-world applications.

Response:

Thank you very much for your attention and constructive feedback on our manuscript.

2. Will the work be of significance to the field and related fields? How does it compare to the established literature? If the work is not original, please provide relevant references.

2.1 VFMGL framework presented in this paper stands out from the established literature in two key aspects. First, VFMGL tackles the challenge of data heterogeneity through its novel DDBL method for selecting low-heterogeneity data and enabling cross-center generalization. Second, while most research has focused on improving model performance, VFMGL achieves substantial computational efficiency through the HGKT transfer learning technique.

2.2 However, it is somewhat uncertain whether the generalization ability was obtained through DDBL's sample selection. To make this clear, ablation studies supporting the effectiveness of DDBL are required.

Response:

Thank you for your feedback, which helps us further refine the results of our paper. In this study, we proposed a VFMGL framework suitable for various tasks and multi-center scenarios, allowing each center to choose local model structures and open-source VFMs based on their data tasks and hardware resources. VFMGL constructs local models in stages: in the first stage, it utilizes the Heterogeneous-model General Knowledge Transfer (HGKT) method to automatically identify and transfer general knowledge applicable to each center's tasks from open-source VFMs, thereby constructing lightweight local models that possess robustness and generalization. Due to the redundancy of parameters in deep learning models [1,2,3], in the second stage, we apply the Data Deduction in Batch Level (DDBL) method to select low-heterogeneity data, enabling the

local model to learn common knowledge from the shared model, further enhancing the local model's cross-center generalization performance.

Based on your suggestions, we have continued to improve our work by adding ablation experiments for the two issues: "3.2.1 Evaluating generalization performance with Stage 1 alone" and "3.2.2 Assessing generalization performance in Stage 2 without DDBL, focusing only on Knowledge Distillation (KD)," and we analyzed the results to validate the performance of VFMGL. In use case 3, the results of 36 cross-center generalization tests show that KD+DDBL helped improve the model's average performance across centers by 0.0429, with a total of 22 generalization results showing significant performance improvement (Fig. 11(e)). In use case 4, the results of 36 cross-center generalization tests indicate that KD+DDBL helped improve the model's average performance across centers by 0.0097, with 25 generalization results showing significant performance improvement (Fig. 12(e)). Ablation experiments demonstrate that the cross-center generalization ability of VFMGL is further enhanced based on the DDBL method.

2.3 Furthermore, in the case of HGKT, the VFM uses a transformer-based model while the local model uses a CNN. However, there is a lack of explanation for why fundamentally different models were chosen and how a linear transformation can effectively address the structural and fundamental differences between these models. We have detailed our concerns in the following questions and hope the authors will address them.

Response:

Thank you very much for your valuable comments!

Medical artificial intelligence holds great potential in assisting with precise cancer diagnosis. However, the data heterogeneity arising from differences in imaging devices, scanning parameters, and image quality across multiple medical centers significantly affects the robustness and generalization of deep learning models (such as ResNet18, UNet, etc.).

Vision foundation models (VFMs) based on transformers, such as DINOv2 and SAM, have a large number of model parameters (ranging from nearly 100M to several billion). These models are trained on large-scale unlabeled natural image datasets using self-supervised methods, learning robust and general visual features that enable them to be applied to various downstream tasks [4,5]. The robust and general visual features of VFM can help mitigate the impact of data heterogeneity on model performance. However, due to the obvious differences between natural and medical images, directly applying these VFM models to medical tasks is challenging [6,7]. Training, fine-tuning, and deploying VFM in the medical field based on existing frameworks require sufficient medical data, hardware resources, and time [6,7]. For instance, Jun Ma et al.[6] fine-tuned SAM to obtain MedSAM using 20 A100 GPUs, each with an 80GB capacity, on a dataset containing over a million medical images. Moreover, due to legislative regulations and the need for medical data security, medical centers are generally not allowed to share data [8,9], making centralized model training difficult. Inspired by these challenges, we conducted this study and proposed a general lightweight framework for vision foundation models (VFMGL) suitable for multi-task and multi-center applications.

Given the significant domain differences between natural and medical images, blindly matching model layers and performing knowledge transfer may not benefit medical tasks and could even lead to negative transfer. Some researchers choose to match and transfer knowledge by fixing intermediate model layers based on experience or experimentation [8,10,11], which is a

time-consuming and labor-intensive task, especially for VFMs with large model layers. As tasks and datasets change, the effectiveness of model layers determined through manual methods remains questionable. Additionally, in multi-center scenarios, the knowledge suitable for local data will vary due to data heterogeneity [9], resulting in differences in the knowledge needed from VFMs by each center.

To address this, we propose the HGKT method, which assists centers in transferring general knowledge from open-source VFMs to local models (Fig.1b). This method adaptively calculates model layer matching weights and feature transfer weights using a meta-network, allowing each center to have customized weights for tailored model layer matching and feature transfer. In this process, we reshape the patch tokens output by the VFM to recover the feature maps of the input images, using bilinear interpolation to align the dimensions of the VFM feature maps with those of the CNN feature maps[17]. Then, we employ a learnable convolutional encoding layer to transform the local model's feature maps to align with the VFM output features, effectively mapping CNN features into a space similar to VFM features for matching. The parameters of this encoding layer are updated only during the training of the local model. Table S15 shows the matching weights for each model layer, indicating substantial differences in the degree of matching between local model layers at different centers and the VFM model layers, corroborating our viewpoint. HGKT automates the matching of model layers and feature transfer between heterogeneous models, saving significant manual effort in multi-task and multi-center scenarios. This process not only achieves lightweight VFM but also enables local models to extract robust features by learning general knowledge. Furthermore, in terms of knowledge transfer functionality, the general applicability of the HGKT method is not limited to Transformer to CNN; it is also suitable for CNN-Transformer, Transformer-Transformer, CNN-CNN scenarios, and more.

3. Does the work support the conclusions and claims, or is additional evidence needed?

3.1 DDBL was proposed to select low-heterogeneity data, but what constitutes "low-heterogeneity data" and its rationale is unclear, it raises several important questions:

3.1.1 Doesn't this method reduce the model's ability to learn diverse features from various samples, thereby potentially limiting its generalization capability?

Response:

Thank you for the valuable comments from the reviewer. The imaging data from multiple medical centers exhibit data heterogeneity due to differences in imaging equipment, scanning parameters, and image quality, which results in bias in the data characteristics across centers. This allows local models to easily fit local data while performing poorly on data from other centers. As neural networks become increasingly powerful, they can learn specific feature patterns from particular datasets to achieve good performance [12,13]. Therefore, we propose the DDBL method, which selects low-heterogeneity data from each center based on shared model knowledge. This approach enables the local model to learn common knowledge from the shared model, thereby suppressing the learning of specific feature patterns by the local model and leveraging the model's redundant parameters to further enhance cross-center generalization capability.

In the VFMGL framework, we construct the local model in two stages. In the first stage, we use the HGKT method to automatically identify and transfer applicable general knowledge from

open-source VFMs for each center's tasks, which helps build a lightweight local model for robust feature extraction. During this stage, we utilize all local data for general knowledge transfer, allowing the local model to stably extract effective features from diverse data. Given the redundancy of deep learning model parameters [1,2,3], this suggests further potential for model performance enhancement. The model layer matching weights computed by the HGKT method reflect the amount of general knowledge learned by the local model layers from the VFMs and the importance of these layers in extracting robust features.

Thus, before proceeding to the second stage of model construction, we freeze the critical layers of the local model that contribute to robustness to prevent the local model from forgetting robust knowledge during the subsequent learning phase. In the federated learning framework, the shared model is aggregated from the local models of multiple centers, integrating knowledge and decision information from these centers. The local model can easily fit data with distinct specific feature expressions on local data, rapidly reducing the discrepancy between the model's predicted distribution and the true label distribution. However, these data are difficult to find similar samples from other centers, resulting in the shared model lacking sufficient knowledge to predict such samples, which exacerbates the differences in prediction distributions between the shared model and the local model. Based on this consideration, we propose the DDBL method, which uses the knowledge of the shared model to select low-heterogeneity data from each center. This, combined with a knowledge distillation approach based on model logic layer outputs, drives the local model to learn common knowledge shared across multiple centers, further enhancing the model's cross-center generalization capability.

3.1.2 Is there a risk that the specific classes or samples might be excluded during this process?

Response:

We thank you for pointing out this important issue.

In the VFMGL framework, the local model is constructed in two phases. In the first phase of local model construction, we use all local data and leverage the HGKT method to transfer general knowledge from the VFM to build the local model. HGKT calculates the model layer matching weights between the VFM model layers and the local model layers, which reflect the amount of general knowledge learned by the local model layers from the VFMs and the criticality of the model layers' robustness. We freeze the model layers in the local model that have a high criticality of robustness to prevent the transferred general knowledge from being forgotten in the next phase.

Furthermore, in the second phase of local model construction, we use the DDBL strategy to select low-heterogeneity data from these datasets for the local model to learn common knowledge from the shared model. The DDBL strategy identifies data in batches and does not specifically exclude any samples. Since each training iteration of the model uses random sampling to obtain data batches, the data batches for each training iteration are not fixed. Therefore, we believe the risk of excluding specific categories or samples is extremely low.

3.1.3 It would be beneficial to see qualitative examples of data classified as low-heterogeneity and high-heterogeneity.

Response:

Thank you for your feedback, which has helped us further refine the study results.

As shown in Fig.S11(A), in the example provided for use case 1, the first column represents examples identified by DDBL as high-heterogeneity data, while the second column shows low-heterogeneity data; the first row contains positive samples, and the second row, negative samples. The figure demonstrates that low-heterogeneity data provide simpler and clearer information useful for diagnosis, with a lower risk of misjudgment. In contrast, high-heterogeneity data offer more complex information, which increases diagnostic difficulty and risk—such as how the intrauterine environment and the smoothness of the inner wall can impact diagnosis. In segmentation tasks (Fig.S11(B)(C)), the segmentation targets in low-heterogeneity images exhibit relatively clear and common boundary information, whereas high-heterogeneity images present segmentation targets with more challenging contours to define, for example, the impact of staining differences on the delineation of nuclear contours.

Fig.S11 high- and low-heterogeneity images

3.2 To verify these concerns and better understand the impact of DDBL, ablation studies are necessary to determine if the cross-center generalization ability was indeed achieved through Stage 2. These studies should include:

3.2.1 Evaluating generalization performance with Stage 1 alone.

Response:

Thank you for your feedback, which helps us further refine the results of our paper. Based on your suggestions, we have added relevant experiments and provided a unified presentation and analysis of the results in section 3.2.2.

3.2.2 Assessing generalization performance in Stage 2 without DDBL, focusing only on Knowledge Distillation (KD).

Response:

In the VFMGL framework, we utilize the HGKT method to transfer general knowledge from the VFM to construct the local model. Due to the inherent characteristics of general knowledge, the local model already exhibits relatively excellent performance at this stage, as validated by the following ablation experiments. In the second phase, we further learn common knowledge from the shared model using KD and DDBL, helping the local model improve cross-center generalization performance.

To validate the above, we conducted ablation experiments on public datasets (use case 3-4). As shown in Fig. 11 and Fig. 12, (a) represents the performance of the model using only HGKT (Stage I), (b) represents the performance of HGKT+KD, and (c) represents the performance of

HGKT+KD+DDBL. To facilitate the comparison of performance changes across different methods, (d) shows the performance change from using only HGKT (a) to HGKT+KD (b), (e) shows the performance change from HGKT (a) to HGKT+KD+DDBL (c), and (f) shows the performance change from HGKT+KD (b) to HGKT+KD+DDBL (c).

In use case 3, KD helped improve the model's average performance across centers by 0.0110, while KD+DDBL increased the model's average performance by 0.0429. Among the 36 cross-center generalization results, KD+DDBL demonstrated significant improvement in model performance in 22 results (Fig.11(e)). In use case 4, KD improved the model's average performance across centers by 0.0068, while KD+DDBL increased the average performance by 0.0097. Among the 36 cross-center generalization results, KD+DDBL showed significant improvement in model performance in 25 results (Fig.12(e)). These comparisons indicate that the performance of the local models on data from other centers can still be improved without compromising their performance on their local data (Fig. 11(e) & Fig. 12(e)).

Fig.11 Ablation experiments of VFMGL in use case 3.

Fig.12 Ablation experiments of VFMGL in use case 4.

3.3 VFMGL's performance on robustness is impressive but could be improved by refining the experimental setup. The current division of the test set into two groups based on the age cutoff of 54.7 is not representative, as endometrial cancer typically affects women around 60. Using a more appropriate cutoff closer to this age would yield more relevant insights. Additionally, considering the higher incidence of endometrial cancer in Black women compared to White women, incorporating ethnicity into the analysis and conducting this experiment if possible would provide a more comprehensive understanding of demographic impacts.

Response:

Thanks for your thoughtful suggestion.

Based on your suggestion, we re-grouped the test set data using an age threshold of 60 and created two groups (as shown in the table below). From the table, it can be observed that patients with no myometrial invasion in each center have an age less than 60, resulting in only positive samples in Group 2 for each center. In Group 1 (Age < 60), the AUC values for each center are 0.771, 0.813, 0.875, and 0.810, respectively. In Group 2 (Age ≥ 60), we calculated the prediction accuracy for each center using the cut-off values from the training set, which are 0.952 (cutoff = 0.7248), 0.714 (cutoff = 0.7659), 0.833 (cutoff = 0.9333), and 0.963 (cutoff = 0.8767). The experimental results indicate that VFMGL maintains robust predictive performance even after data re-grouping.

As you pointed out, the incidence of endometrial cancer is indeed influenced by ethnicity differences. However, the EC dataset used in this study comprises only Chinese patients, making it difficult to incorporate ethnicity into the analysis. Global collaboration would greatly facilitate achieving this research objective, which is also a prospect we hope to pursue in the future.

TestSet	Group	Center A		Center B		Center C		Center D	
		NMI	MI	NMI	MI	NMI	MI	NMI	MI
	Group1(Age<60)	34	179	6	16	1	8	8	83
	Group2(Age≥60)	0	62	0	7	0	6	0	27

Notes: MI, myometrial invasion; NMI, non myometrial invasion.

3.4 Furthermore, since the authors used a private dataset for ‘VFMGL exhibits robustness’ experiments, conducting experiments using a public dataset would allow for future comparisons with other studies. If demographic information is not available for the public dataset, which makes part 1 of the experiment infeasible, conducting part 2 of the experiment would still be valuable.

Response:

Thank you for your suggestions, which have helped us further improve the results of our paper.

To allow for future comparisons with other studies, we have further improved our work based on your suggestions by adding random permutation experiments on three public datasets (use case 2-4). To ensure fairness, the hyperparameter settings are consistent with the initial experiment setup (as reported in the manuscript). The distribution of each dataset is shown in Tables S9-S11.

In the Breast Cancer Histology Image Dataset (use case 2), the robustness performance of VFMGL across six different data distributions is shown in Fig.S8, with the AUC performance variations for each center being: 0.9990±0.0001 (Center A), 0.9978±0.0003 (Center B), 0.9988±0.0004 (Center C), 0.9966±0.0008 (Center D), and 0.9989±0.0002 (Center E).

In the Prostate MRI Dataset (use case 3), the robustness performance of VFMGL is shown in Fig.S9, with Dice performance variations for each center as follows: 0.9134 ± 0.0121 , 0.9258 ± 0.0098 , 0.9546 ± 0.0060 , 0.9297 ± 0.0100 , 0.9246 ± 0.0106 , and 0.9013 ± 0.0072 . In the Histology Nuclei Dataset (use case 4), the robustness performance of VFMGL is shown in Fig.S10, with Dice performance variations for each center as follows: 0.7665 ± 0.0203 , 0.7652 ± 0.0105 , 0.7823 ± 0.0187 , 0.7710 ± 0.0160 , 0.7562 ± 0.0095 , and 0.8233 ± 0.0209 .

The extensive experiments (both classification and segmentation) indicate that VFMGL maintains excellent robustness performance in the face of various data distribution shifts. More detailed robustness performance results of VFMGL on these three datasets can be found in Tables S12-S14.

Fig.S8 The experimental results of VFMGL in use case 2.

Fig.S9 The experimental results of VFMGL in use case 3.

Fig.S10 The experimental results of VFMGL in use case 4.

Table S9. Multiple Permutation Distribution for Use Case 2

Distribution	Random Seed	partition ratio	Set	Center				
				A	B	C	D	E
initial			Train Set	38039	22339	54435	83096	93902
			Val Set	9509	5584	13608	20774	23475
			Test Set	11888	6981	17011	25968	29345
1	50	0.50	Train Set	23775	13962	34023	51936	58690
			Val Set	5943	3490	8505	12984	14672
			Test Set	29718	17452	42526	64918	73360
2	100	0.55	Train Set	26152	15359	37424	57130	64559
			Val Set	6538	3839	9356	14282	16139
			Test Set	26746	15706	38274	58426	66024
3	200	0.65	Train Set	30908	18151	44229	67517	76296
			Val Set	7726	4537	11057	16879	19074
			Test Set	20802	12216	29768	45442	51352
4	500	0.70	Train Set	33285	19548	47631	72711	82165
			Val Set	8321	4886	11907	18177	20541
			Test Set	17830	10470	25516	38950	44016
5	1000	0.80	Train Set	38040	22340	54436	83098	93903
			Val Set	9510	5584	13608	20774	23475
			Test Set	11886	6980	17010	25966	29344

Table S10. Multiple Permutation Distribution for Use Case 3

Distribution	Random Seed	partition ratio	Set	Center					
				A	B	C	D	E	F
initial			Train Set	167	100	299	263	245	112
			Val Set	41	25	75	65	62	28
			Test Set	53	32	94	83	77	35
1	50	0.50	Train Set	104	62	187	164	153	70
			Val Set	26	16	47	41	39	17
			Test Set	131	79	234	206	192	88
2	100	0.55	Train Set	114	69	205	180	168	77
			Val Set	29	17	52	46	43	19
			Test Set	118	71	211	185	173	79
3	200	0.65	Train Set	135	81	243	213	199	91
			Val Set	34	21	61	54	50	22
			Test Set	92	55	164	144	135	62
4	500	0.70	Train Set	146	87	262	230	215	98
			Val Set	36	22	65	57	53	24
			Test Set	79	48	141	124	116	53
5	1000	0.80	Train Set	167	100	299	263	245	112
			Val Set	41	25	75	65	62	28
			Test Set	53	32	94	83	77	35

Table S11. Multiple Permutation Distribution for Use Case 4

Distribution	Random Seed	partition ratio	Set	Center					
				A	B	C	D	E	F
initial			Train Set	47	31	44	46	24	26
			Val Set	11	8	11	11	6	7
			Test Set	22	22	31	26	14	17
1	50	0.50	Train Set	32	24	34	32	17	20
			Val Set	8	6	9	9	5	5
			Test Set	40	31	43	42	22	25
2	100	0.55	Train Set	35	26	37	36	19	21
			Val Set	9	7	10	9	5	6
			Test Set	36	28	39	38	20	23
3	200	0.65	Train Set	41	31	44	42	22	25
			Val Set	11	8	11	11	6	7
			Test Set	28	22	31	30	16	18
4	500	0.70	Train Set	44	33	48	46	24	28
			Val Set	12	9	12	12	6	7
			Test Set	24	19	26	25	14	15
5	1000	0.80	Train Set	51	38	54	52	28	32

Val Set	13	10	14	14	7	8
Test Set	16	13	18	17	9	10

Table S12. The experimental results of VFMGL under various data distributions in use case 2.

Center	Methods	AUC	Sensitivity	Specificity	Accuracy	PPV	NPV	F1
A	initial	0.9992	0.9882 (5874/5944)	0.9896 (5882/5944)	0.9889 (11756/11888)	0.9896 (5874/5936)	0.9882 (5882/5952)	0.9889
	1	0.9989	0.9896 (14704/14859)	0.9860 (14651/14859)	0.9878 (29355/29718)	0.9861 (14704/14912)	0.9895 (14651/14806)	0.9878
	2	0.9988	0.9765 (13059/13373)	0.9951 (13308/13373)	0.9858 (26367/26746)	0.9950 (13059/13124)	0.9769 (13308/13622)	0.9857
	3	0.9991	0.9863 (10259/10401)	0.9894 (10291/10401)	0.9879 (20550/20802)	0.9894 (10259/10369)	0.9864 (10291/10433)	0.9879
	4	0.9991	0.9874 (8803/8915)	0.9920 (8844/8915)	0.9897 (17647/17830)	0.9920 (8803/8874)	0.9875 (8844/8956)	0.9897
	5	0.9990	0.9837 (5846/5943)	0.9909 (5889/5943)	0.9873 (11735/11886)	0.9908 (5846/5900)	0.9838 (5889/5986)	0.9872
B	initial	0.9973	0.9481 (3309/3490)	0.9974 (3482/3491)	0.9728 (6791/6981)	0.9973 (3309/3318)	0.9506 (3482/3663)	0.9721
	1	0.9977	0.9589 (8367/8726)	0.9936 (8670/8726)	0.9762 (17037/17452)	0.9934 (8367/8423)	0.9602 (8670/9029)	0.9758
	2	0.9978	0.9826 (7716/7853)	0.9808 (7702/7853)	0.9817 (15418/15706)	0.9808 (7716/7867)	0.9825 (7702/7839)	0.9817
	3	0.9980	0.9751 (5956/6108)	0.9903 (6049/6108)	0.9827 (12005/12216)	0.9902 (5956/6015)	0.9755 (6049/6201)	0.9826
	4	0.9981	0.9866 (5165/5235)	0.9811 (5136/5235)	0.9839 (10301/10470)	0.9812 (5165/5264)	0.9866 (5136/5206)	0.9839
	5	0.9977	0.9731 (3396/3490)	0.9911 (3459/3490)	0.9821 (6855/6980)	0.9910 (3396/3427)	0.9735 (3459/3553)	0.9819
C	initial	0.9995	0.9911 (8430/8506)	0.9915 (8433/8505)	0.9913 (16863/17011)	0.9915 (8430/8502)	0.9911 (8433/8509)	0.9913
	1	0.9985	0.9856 (20956/21263)	0.9843 (20929/21263)	0.9849 (41885/42526)	0.9843 (20956/21290)	0.9855 (20929/21236)	0.9849
	2	0.9989	0.9718 (18598/19137)	0.9948 (19037/19137)	0.9833 (37635/38274)	0.9947 (18598/18698)	0.9725 (19037/19576)	0.9831
	3	0.9989	0.9765 (14534/14884)	0.9942 (14797/14884)	0.9853 (29331/29768)	0.9940 (14534/14621)	0.9769 (14797/15147)	0.9852
	4	0.9985	0.9911 (12644/12758)	0.9676 (12345/12758)	0.9793 (24989/25516)	0.9684 (12644/13057)	0.9908 (12345/12459)	0.9796
	5	0.9987	0.9780 (8318/8505)	0.9937 (8451/8505)	0.9858 (16769/17010)	0.9935 (8318/8372)	0.9784 (8451/8638)	0.9857
D	initial	0.9977	0.9427 (12240/12984)	0.9988 (12969/12984)	0.9708 (25209/25968)	0.9988 (12240/12255)	0.9457 (12969/13713)	0.9699
	1	0.9957	0.9607 (31183/32459)	0.9819 (31870/32459)	0.9713 (63053/64918)	0.9815 (31183/31772)	0.9615 (31870/33146)	0.9710
	2	0.9968	0.9516 (27800/29213)	0.9940 (29039/29213)	0.9728 (56839/58426)	0.9938 (27800/27974)	0.9536 (29039/30452)	0.9722
	3	0.9970	0.9682 (21999/22721)	0.9883 (22456/22721)	0.9783 (44455/45442)	0.9881 (21999/22264)	0.9688 (22456/23178)	0.9781
	4	0.9956	0.9625 (18744/19475)	0.9820 (19124/19475)	0.9722 (37868/38950)	0.9816 (18744/19095)	0.9632 (19124/19855)	0.9719
	5	0.9967	0.9670 (12554/12983)	0.9879 (12826/12983)	0.9774 (25380/25966)	0.9876 (12554/12711)	0.9676 (12826/13255)	0.9772
E	initial	0.9993	0.9845 (14445/14672)	0.9922 (14559/14673)	0.9884 (29004/29345)	0.9922 (14445/14559)	0.9846 (14559/14786)	0.9883
	1	0.9986	0.9853 (36141/36680)	0.9849 (36126/36680)	0.9851 (72267/73360)	0.9849 (36141/36695)	0.9853 (36126/36665)	0.9851
	2	0.9987	0.9915 (32730/33012)	0.9798 (32345/33012)	0.9856 (65075/66024)	0.9800 (32730/33397)	0.9914 (32345/32627)	0.9857
	3	0.9989	0.9837 (25257/25676)	0.9921 (25472/25676)	0.9879 (50729/51352)	0.9920 (25257/25461)	0.9838 (25472/25891)	0.9878
	4	0.9989	0.9913 (21816/22008)	0.9852 (21683/22008)	0.9883 (43499/44016)	0.9853 (21816/22141)	0.9912 (21683/21875)	0.9883
	5	0.9990	0.9913 (14544/14672)	0.9851 (14453/14672)	0.9882 (28997/29344)	0.9852 (14544/14763)	0.9912 (14453/14581)	0.9882

Notes: AUC, area under the curve; PPV, positive predictive value; NPV, negative predictive value.

Table S13. The experimental results of VFMGL under various data distributions in use case 3.

Center	Methods	Dice	ASSD	Sensitivity	Specificity	PPV	IOU	F1
A	initial	0.9340	5.6570	0.8981	0.9953	0.8685	0.7872	0.8831
	1	0.9005	8.5498	0.9074	0.9892	0.7638	0.7060	0.8294
	2	0.9040	9.7341	0.9079	0.9905	0.7904	0.7240	0.8451
	3	0.9196	8.0357	0.8951	0.9926	0.8180	0.7420	0.8548
	4	0.9136	7.8488	0.9155	0.9909	0.7823	0.7214	0.8437
	5	0.9089	10.0914	0.9047	0.9921	0.7736	0.7083	0.8340
B	initial	0.9328	5.0626	0.9337	0.9956	0.8196	0.7659	0.8729
	1	0.9115	6.8731	0.9129	0.9950	0.7937	0.7339	0.8492
	2	0.9258	4.8663	0.9196	0.9960	0.8194	0.7525	0.8666
	3	0.9304	5.5225	0.9043	0.9965	0.8305	0.7601	0.8658
	4	0.9170	4.9035	0.9139	0.9950	0.7983	0.7443	0.8522
	5	0.9371	3.4074	0.9290	0.9972	0.8418	0.7845	0.8832
C	initial	0.9620	2.8192	0.9539	0.9973	0.9076	0.8672	0.9302
	1	0.9485	3.8772	0.9271	0.9970	0.8897	0.8270	0.9080
	2	0.9569	3.5448	0.9298	0.9972	0.9013	0.8421	0.9153
	3	0.9543	3.5278	0.9291	0.9969	0.8956	0.8377	0.9120
	4	0.9466	4.6275	0.9424	0.9955	0.8563	0.8100	0.8973
	5	0.9592	2.5650	0.9381	0.9978	0.9246	0.8695	0.9313
D	initial	0.9191	9.4369	0.8434	0.9966	0.8370	0.7465	0.8402
	1	0.9281	7.4399	0.9150	0.9947	0.8320	0.7788	0.8715
	2	0.9255	7.2161	0.9125	0.9956	0.8327	0.7777	0.8708
	3	0.9334	6.8064	0.9198	0.9946	0.8150	0.7627	0.8642
	4	0.9476	6.0605	0.9210	0.9957	0.8426	0.7822	0.8800
	5	0.9243	9.7444	0.9096	0.9943	0.8044	0.7509	0.8538
E	initial	0.9383	6.0202	0.9296	0.9946	0.8501	0.8028	0.8881
	1	0.9131	9.6877	0.9203	0.9913	0.8024	0.7439	0.8573
	2	0.9308	6.9545	0.9291	0.9915	0.8188	0.7703	0.8705
	3	0.9166	9.4326	0.9038	0.9896	0.7892	0.7457	0.8426
	4	0.9159	8.4609	0.9331	0.9900	0.7800	0.7418	0.8497
	5	0.9329	5.1608	0.9026	0.9949	0.8570	0.7879	0.8792
F	initial	0.9080	6.8139	0.8981	0.9942	0.7973	0.7445	0.8447
	1	0.9022	6.5187	0.8782	0.9957	0.8097	0.7134	0.8425
	2	0.8932	7.9375	0.8689	0.9939	0.7801	0.7069	0.8221
	3	0.9083	8.1140	0.8775	0.9956	0.8148	0.7402	0.8450
	4	0.9041	7.3885	0.9216	0.9932	0.7722	0.7246	0.8403
	5	0.8918	7.7375	0.8909	0.9953	0.7952	0.7166	0.8403

Notes: ASSD, average symmetric surface distance; PPV, positive predictive value; IOU: intersection over union.

Table S14. The experimental results of VFMGL under various data distributions in use case 4.

Center	Distribution	Dice	ASSD	Sensitivity	Specificity	PPV	IOU	F1
A	Initial	0.7509	3.6429	0.5774	0.9606	0.8257	0.4918	0.6796
	1	0.7574	6.6052	0.6516	0.9404	0.6632	0.4406	0.6573
	2	0.7955	6.4820	0.7688	0.9170	0.6340	0.5054	0.6949
	3	0.7820	6.1335	0.7001	0.9318	0.6701	0.4895	0.6848
	4	0.7715	6.1798	0.7149	0.9286	0.6423	0.4700	0.6766
	5	0.7415	6.7174	0.6362	0.9346	0.6238	0.4323	0.6300
B	Initial	0.7658	4.1422	0.6245	0.9347	0.7240	0.4731	0.6706
	1	0.7717	4.6163	0.6674	0.9366	0.6840	0.4777	0.6756
	2	0.7816	4.7190	0.6996	0.9156	0.6239	0.4653	0.6596
	3	0.7559	7.7385	0.7909	0.8771	0.5099	0.4388	0.6201
	4	0.7528	4.9399	0.6696	0.9221	0.6399	0.4586	0.6545
	5	0.7632	7.0551	0.7379	0.9100	0.5853	0.4341	0.6528
C	Initial	0.7735	8.2582	0.6203	0.9646	0.7653	0.4823	0.6852
	1	0.7700	9.7468	0.7477	0.9253	0.5885	0.4502	0.6586
	2	0.7703	10.6894	0.7357	0.9227	0.6009	0.4593	0.6615
	3	0.7733	10.9627	0.7115	0.9201	0.5707	0.4314	0.6334
	4	0.8176	5.8284	0.7548	0.9307	0.6793	0.5251	0.7151
	5	0.7893	9.0939	0.7107	0.9484	0.6529	0.4861	0.6806
D	Initial	0.7410	3.6780	0.5059	0.9681	0.8440	0.4471	0.6326
	1	0.7793	6.4168	0.7545	0.9068	0.6190	0.4808	0.6801
	2	0.7825	5.6679	0.7487	0.9064	0.6292	0.4945	0.6838
	3	0.7794	4.8177	0.6492	0.9390	0.7212	0.4926	0.6833
	4	0.7647	5.7593	0.6518	0.9286	0.6955	0.4647	0.6729
	5	0.7794	5.2083	0.6974	0.9207	0.6746	0.5014	0.6858
E	Initial	0.7568	1.0707	0.6386	0.8806	0.6821	0.4914	0.6596
	1	0.7552	1.1665	0.6574	0.8775	0.6549	0.4851	0.6562
	2	0.7506	1.1697	0.6480	0.8822	0.6840	0.4938	0.6655
	3	0.7746	1.0311	0.6804	0.8951	0.7071	0.5265	0.6935
	4	0.7492	1.0805	0.6696	0.8875	0.6733	0.4946	0.6714
	5	0.7509	1.2700	0.6420	0.8854	0.6368	0.4557	0.6394
F	Initial	0.7899	4.1967	0.6218	0.9729	0.7339	0.4772	0.6732
	1	0.8276	3.9756	0.7545	0.9645	0.7097	0.5415	0.7314
	2	0.8065	6.1754	0.8185	0.9563	0.6114	0.4968	0.7000
	3	0.8357	3.7802	0.7583	0.9692	0.7242	0.5574	0.7408
	4	0.8446	4.2443	0.7073	0.9753	0.7540	0.5446	0.7299
	5	0.8358	8.2434	0.7934	0.9725	0.6613	0.5383	0.7213

Notes: ASSD, average symmetric surface distance; PPV, positive predictive value; IOU: intersection over union.

4. Are there any flaws in the data analysis, interpretation and conclusions? Do these prohibit publication or require revision?

4.1 N/A

5. Is the methodology sound? Does the work meet the expected standards in your field?

5.1 In the DDBL process, outlier samples are excluded to maintain low heterogeneity. Given that only features corresponding to a similar data distribution are selected, what is the rationale for claiming this process contributes to generalizability?

Response:

Thank you for the valuable comments from the reviewer.

Due to variations in imaging devices, scanning parameters, and image quality, multi-center medical imaging data exhibit data heterogeneity, causing discrepancies in data characteristics across centers. This often allows local models to fit easily on local data but perform poorly on data from other centers. As neural networks become increasingly powerful, they can learn specific feature patterns within certain datasets to achieve high performance[12,13]. In a federated learning framework, the shared model is aggregated from local models across multiple centers, integrating knowledge and decision-making information from these centers[9,14,15]. Local models can quickly reduce the discrepancy between the prediction distribution and the true label distribution on local data with pronounced specific features. However, such data may lack similar samples in other centers, resulting in the shared model lacking sufficient knowledge to predict these types of samples and increasing the prediction distribution discrepancy between the shared and local models.

In light of this, we propose the DDBL method, which selects low-heterogeneity data from each center based on the shared model's knowledge. By employing Knowledge Distillation (KD) based on model logic layer outputs, the method drives local models to learn common knowledge from the shared model, thereby suppressing the learning of specific feature patterns in local models and utilizing model redundancy to enhance cross-center generalization capability. Additionally, within the VFMGL framework, we construct local models in two stages. In the first stage, we use all local data for general knowledge transfer, enabling the local models to stably extract effective features from diverse data. Before constructing models in the second stage, we freeze robustness critical layers in the local models to prevent them from forgetting robust knowledge during further learning.

5.2 Why does VFM use a transformer-based model while the local model uses a CNN? Despite the differences in the learning mechanisms between the two models, there seems to be some mismatch. An explanation is needed on why the weight transformer could still be effective in this context.

Response:

We thank you for pointing out this important issue.

In recent years, transformer-based VFMs, such as DINOv2 and SAM, have shown great potential in the image domain. VFMs are trained on large-scale data with extensive model parameters, utilizing self-attention mechanisms to enhance feature learning capability, enabling more general and robust feature representations. However, VFMs in the medical field still face challenges related to data scale and computational resources. In a multi-center medical setting, some clients may lack sufficient resources and infrastructure to train and deploy these models,

making it more suitable to choose CNN networks, which have lower computational complexity and fewer parameters, as local models. Currently, CNNs remain widely used in the visual field and include many classic architectures (e.g., VGG, ResNet, Inception). Notably, the versatility of the HGKT method is not limited to Transformer-to-CNN knowledge transfer; it also applies to knowledge transfer from VFMs to transformer models. In this study, we chose the widely recognized ResNet18 and UNet as local models to demonstrate the practicality and generalizability of VFMGL.

Transformers divide images into fixed-size patches, encode and flatten them into patch tokens, and utilize self-attention mechanisms to enable interaction among patch tokens, thereby representing image features. CNNs, on the other hand, extract local features layer by layer through convolution operations, gradually expanding the receptive field to capture global information, representing image features through feature maps. Although these two model structures and learning mechanisms differ, the output features of their encoding layers have similar representations[16], making layer matching and knowledge transfer feasible.

We customized meta-networks for knowledge transfer in each center, leveraging these networks to automatically compute layer matching weights and feature transfer weights, enabling each center to have tailored weights for customized layer matching and feature transfer. In this process, we reshape the patch tokens output by the VFM to restore the feature map of the input image[17]. By using bilinear interpolation, we align the VFM feature map with the CNN feature map in size, while using a learnable convolutional encoding layer to transform the local model's features for alignment with VFM output features. This effectively maps CNN features to a space similar to VFM features. The parameters of this encoding layer are updated only during the training of local models. Based on this mechanism, there is no need to convert the model parameter weights between the two models. Instead, the process of local model learning from general knowledge is achieved by calculating the representation difference between VFM and the local model, and constructing a knowledge transfer loss that combines model layer matching weights and feature transfer weights. The model layer matching weights measure the transferability between VFM and the local model at the model layer level, while the feature transfer weights are used to select the appropriate knowledge for transfer. In summary, through the HGKT approach, we successfully achieved the adaptive acquisition of general knowledge from VFM, applicable to local medical tasks.

5.3 The authors emphasize the computational benefits of the lightweight local models in VFMGL. However, the overall process requires a complex training procedure, resulting in substantial computational costs. Notably, the final outcome is a separate local model for each medical center and task, resulting in a total model count of (# of centers) x (# of tasks). Although the individual models are lightweight, the cumulative cost of constructing models is still high. The authors should address this limitation and discuss strategies to reduce the overall model construction overhead or examine the tradeoffs between individual model efficiency and total modeling costs.

Response:

Thank you for the valuable comments from the reviewer.

Your consideration holds when each center has the same tasks. It should be noted that in a multi-center scenario, we are more likely to encounter situations where only some centers share the same task. For example, Centers A, B, D, and F may have the same task, while Centers B, C,

D, E, and F collaborate on other tasks. Thus, the total number of models is not always the product of (number of centers) x (number of tasks). Moreover, by evaluating the similarity in data features between the new center's data and data from other centers within the VFMGL framework, we can select suitable local models for predictions on new center data, which helps alleviate the model-building costs of continually adding new centers to the framework. We can determine whether to include a new center in the framework by measuring the similarity of data features between centers, The threshold setting for data feature similarity between centers needs to balance the model construction cost and model performance.

In use case 1, the multi-center data we used encompasses hospitals at multiple levels (with differences in region, hospital scale, and population served scale). The local models built through the VFMGL framework can be referenced and utilized by other new centers with similar conditions. This approach provides valuable insights and guidance for building collaborative models across a broader regional scope in the future.

6. Is there enough detail provided in the methods for the work to be reproduced?

6.1 Page 5, lines 122 to 123: How was the similarity of data distribution between data centers measured?

Response:

Thanks for your thoughtful suggestion

To calculate inter-center data features similarity, we use each center's local model to extract features from both the local training dataset and the external validation center dataset. Then, we normalize the external validation dataset's features based on the mean and variance of each center's training dataset features. We compute the cosine similarity of the data features between centers using the `cosine_similarity` function from the scikit-learn library. The averaged similarity score is used to measure the similarity in data features between centers.

6.2 What do the values on the x and y axes of the heatmap in Fig.10 represent? If the alphabet represents 'data center,' do the numbers following it refer to samples or to one of the features among the 256 features?

Response:

Thank you for the valuable comments from the reviewer. The alphabet represents data centers, and the numbers represent one of the 256 features. For feature analysis, we use each center's local model to extract features from all centers' test and training data. We then perform a round of Maximum Relevance Minimum Redundancy (mRMR) feature selection on the features extracted by each model, ultimately identifying 256 features to construct a feature correlation heatmap (for more details, see the "Exploration of adaptive knowledge and common knowledge" section in the Methods of the main text and Supplementary S6).

6.3 In the supplementary materials, what do θ_{wfm} , L_{wfm} , and L_{org} mean? Does wfm and org refer to the local model?

Response:

Thanks for your thoughtful suggestion.

Based on your feedback, we carefully reviewed the supplementary materials and found that the symbol θ_{wfm} does not appear to be present. Therefore, we believe you may be referring to θ , which represents the local model parameters, and θ_k represents the local model of the k-th center. L_{wfm} is the feature transfer constraint function for VFM to the local model, while L_{org} is the original loss (standard cross-entropy loss for classification tasks and Dice loss for segmentation tasks). 'wfm' stands for weighted feature matching, and 'org' stands for the original loss (in this study, standard cross-entropy loss is used for classification tasks and Dice loss is used for segmentation tasks).

6.4 The paper lacks training details (e.g., which models were used for open-source VFM, shared model, and local models, batch size, iteration steps, and other hyper-parameters).

Response:

Thank you for your suggestions to help us further improve the paper.

Based on your feedback and advice, We have added the experimental setup details in the first part of the methods section, including the specific models used for the open-source VFM, shared model, and local model, as well as batch size, iteration steps, and other hyperparameters.

“Experimental Setting. We use DINOv2 as the open-source visual foundation model (VFM) and adopt two commonly used deep learning frameworks, ResNet18 and UNet, as local and shared models for classification tasks (use case 1-2) and segmentation tasks (use case 3-4), respectively. In use case 1, we use standard cross-entropy loss and an SGD optimizer to update the local model. The model is trained for 20 epochs with a learning rate of 0.0001, weight decay of 0.0001, and a batch size of 25. In use case 2, we use standard cross-entropy loss and an SGD optimizer to update the local model. The model is trained for 10 epochs with a learning rate of 0.01, weight decay of 0.001, and a batch size of 32. In use case 3, we use Dice loss and an Adam optimizer to update the local model. The model is trained for 100 epochs with a learning rate of 0.0001, weight decay of 0.0001, and a batch size of 3. In use case 4, we use Dice loss and an SGD optimizer to update the local model. The model is trained for 100 epochs with a learning rate of 0.001, weight decay of 0.0001, and a batch size of 2. For all four tasks, the optimizer momentum is set to 0.9, and the meta-network parameters are updated using an SGD optimizer with a learning rate and weight decay of 0.001. Local model training is performed once per communication round.”

6.4.1 Specifically, regarding the weight transfer strategy, what VFM model was used? The supplementary table S7 notes that DINO-V2 was used as the source model and ResNet-18 as the expert model for clinical diagnosis in use case 1. Is this setting fixed?

Response:

Thank you for the valuable comments from the reviewer

The VFM models mentioned in this study are all based on the DINOv2 framework. In the four reported tasks (classification and segmentation), DINOv2 was used as the source model, while ResNet-18 and UNet were used as the expert models for classification and segmentation tasks, respectively. The specific settings for each task have been supplemented in the experimental

setting details.

From the perspective of knowledge transfer, the generality of the HGKT method is not limited to transformer-to-CNN transfers; it is also applicable for transferring knowledge from VFM to lightweight transformer models. In this study, we have chosen the widely-used ResNet18 and UNet as local models to demonstrate the practicality and versatility of the VFMGL framework.

6.4.2 When performing classification and segmentation, is the same VFM model used for both tasks?

Response:

Thank you for the valuable comments from the reviewer

Currently, in this study, we have only explored the use of DINOv2 as the VFM model. DINOv2 is a high-performance foundational computer vision model, whose rich visual general knowledge is suitable for classification, segmentation, and other tasks. This transfer strategy based on the HGKT method loosens the constraints on VFM selection, allowing each center to choose an appropriate VFM according to its own circumstances (data scale, hardware resources, etc.).

6.4.3 For segmentation, on which model is it based? If ResNet-18 is used as the encoder, what model is used as the decoder?

Response:

Thanks for your thoughtful suggestion. In this study, the segmentation tasks are currently all implemented using the UNet model as the local model. The UNet network is a classic model that is widely used. Additionally, as previously mentioned, the framework's versatility is not limited to Transformer-to-CNN transfers but also supports knowledge transfer from VFM to lightweight transformer models.

References

- [1] Geoffrey Hinton, Oriol Vinyals, and Jeff Dean. Distilling the knowledge in a neural network. NIPS 2014.
- [2] Song Han, Huizi Mao, and William J Dally. Deep compression: Compressing deep neural networks with pruning, trained quantization and huffman coding. ICLR 2016.
- [3] K. Zhu, X. Hu, J. Wang, et al. Improving Generalization of Adversarial Training via Robust Critical Fine-Tuning. 2023 IEEE/CVF International Conference on Computer Vision (ICCV). 4401-4411 (2023).
- [4] Kirillov A , Mintun E , Ravi N ,et al. Segment Anything. 2023 IEEE/CVF International Conference on Computer Vision (ICCV). 3992-4003 (2023).
- [5] Maxime Oquab, Timothée Darcet, Théo Moutakanni. et al. DINOv2: Learning Robust Visual Features without Supervision. Preprint at <https://arxiv.org/pdf/2304.07193> (2023)
- [6] Ma, J., He, Y., Li, F. et al. Segment anything in medical images. Nat Commun 15, 654 (2024). <https://doi.org/10.1038/s41467-024-44824-z>
- [7] Vorontsov, E., Bozkurt, A., Casson, A. et al. A foundation model for clinical-grade computational pathology and rare cancers detection. Nat Med 30, 2924–2935 (2024). <https://doi.org/10.1038/s41591-024-03141-0>
- [8] Chen YQ, Lu W, Qin X, Wang JD and Xie X. MetaFed: Federated Learning among

Federations with Cyclic Knowledge Distillation for Personalized Healthcare. IJCAI'22 federated learning workshop.

[9] Jiang MR, Wang ZR, Dou Q. HarmoFL: Harmonizing Local and Global Drifts in Federated Learning on Heterogeneous Medical Images. Proceedings of the 36th AAAI Conference on Artificial Intelligence. **36**, 914-922 (2022).

[10] Romero, A., Ballas, N., Kahou, S. E., Chassang, A., Gatta, C., & Bengio, Y. (2015). Fitnets: Hints for thin deep nets. In ICLR.

[11] Wang X , Fu T , Liao S ,et al.Exclusivity-Consistency Regularized Knowledge Distillation for Face Recognition[C]//European Conference on Computer Vision. Springer, Cham, 2020.DOI:10.1007/978-3-030-58586-0_20.

[12] Antonio Torralba and Alexei A Efros. Unbiased look at dataset bias. CVPR 2011.

[13] Zhuang Liu, Kaiming He. A Decade's Battle on Dataset Bias: Are We There Yet? <https://doi.org/10.48550/arXiv.2403.08632>

[14] McMahan HB, Moore E, Ramage D, Hampson S, Arcas B. Communication-efficient learning of deep networks from decentralized data. 20th International Conference on Artificial Intelligence and Statistics (AISTATS). **54**, 1273-1282 (2016).

[15] Li, Qinbin, Bingsheng He, and Dawn Song. Model-contrastive federated learning. Proceedings of the IEEE/CVF conference on computer vision and pattern recognition. 2021.

[16] Raghu, Maithra, et al. "Do vision transformers see like convolutional neural networks?." Advances in neural information processing systems 34 (2021): 12116-12128.

[17] Leem S, Seo H. Attention Guided CAM: Visual Explanations of Vision Transformer Guided by Self-Attention[C]//Proceedings of the AAAI Conference on Artificial Intelligence. 2024, 38(4): 2956-2964.

Reviewer #3 (Remarks to the Author):

This paper addresses the efficiency of deep foundational models for medical image diagnosis. It primarily focuses on distilling open-sourced vision transformers (ViTs) and transferring the distilled knowledge to local convolutional neural networks (CNNs). Additionally, this work explores topics such as federated learning to ensure model training on patient-sensitive data. However, in my opinion, mechanically combining both model distillation and federated learning is neither novel nor particularly useful.

Response:

Thank you very much for your attention and constructive feedback on our manuscript.

- The concept of distilling CNNs from ViTs has been extensively researched since 2022 [Yao, Xufeng, et al. "Distill Vision Transformers to CNNs via Low-Rank Representation Approximation."]. Furthermore, similar and more intuitive research on distilling a CNN to train a ViT better has been thoroughly investigated by the famous DeiT paper. These previous works significantly challenge the novelty of this paper.

Response:

Thank you for your suggestions. It seems that our previous explanation may have caused some confusion. Allow me to provide a detailed description of our main work here.

Medical AI has substantial potential to aid in the precise diagnosis of cancer. However, the heterogeneity of imaging data across multiple medical centers, caused by differences in imaging equipment, scanning parameters, and image quality, greatly affects the robustness and generalization of deep learning models (such as ResNet18 and UNet).

Unlike other DL models, vision foundation models (VFMs), such as DINOv2 and SAM, have extensive model parameters (ranging from approximately 100 million to several billion). These VFMs are trained using self-supervised methods on massive, unlabeled natural image datasets, enabling them to learn robust and generalizable visual features that make VFMs applicable to various downstream tasks [1,2]. The robust, generalized visual features of VFMs can help reduce the impact of data heterogeneity on model performance. However, natural images differ significantly from medical images, and directly applying the aforementioned VFMs to medical tasks often fails to meet the requirements [3,4]. Training, fine-tuning, and deploying VFMs specifically for medical applications require ample medical data, hardware resources, and time [3,4]. For instance, Jun Ma et al. fine-tuned SAM on a dataset of over one million medical images using 20 A100 GPUs (each with 80GB of memory) to develop MedSAM [3]. Furthermore, due to legislative requirements and the need for medical data security, data sharing across medical centers is generally prohibited [5,6], making centralized model training infeasible.

Inspired by the above, we conducted this study and proposed a Vision Foundation Model General Lightweight (VFMGL) framework, designed for multi-task, multi-center scenarios. VFMGL constructs local models in two stages. In the first stage, the Heterogeneous-model General Knowledge Transfer (HGKT) method is used to automatically identify and transfer general knowledge from open-source VFMs, suited to each center's tasks, to create a lightweight local model with robustness and generalizability. Given that deep learning models have parameter redundancy [7,8,9], in the second stage, we employ the Data Deduction in Batch Level (DDBL) method to drive local models to learn common knowledge from a shared model using

low-heterogeneity data, further improving the cross-center generalization of local models. The specific contributions and advantages are as follows:

Firstly, Given the significant domain differences between natural and medical images, arbitrarily matching model layers for knowledge transfer may not benefit, and could even hinder, medical tasks. Some researchers rely on empirical or experimental selection of fixed intermediate layers for matching and knowledge transfer [5,10,11], a time-consuming task, particularly for VFMs with large model layers. With task and dataset changes, the effectiveness of layer-matching based on manual methods remains uncertain. Moreover, in a multi-center scenario, data heterogeneity can lead to variation in the knowledge needed for each center's local model [6]. Accordingly, the type of knowledge each center requires from VFMs can vary.

To address this, we propose the HGKT method to assist each center in transferring general knowledge from open-source VFMs to local models (Fig.1b). We use a meta-network to adaptively calculate the layer-matching weights and feature transfer weights, allowing each center to have customized weights for layer matching and feature transfer. Table S15 presents the matching weights for different model layers, showing significant differences in the matching between local model layers at different centers and VFM layers, which supports the above perspective. HGKT automates layer matching and feature transfer across heterogeneous models in multi-task, multi-center scenarios, saving significant manual effort. At this stage, the local model leverages the HGKT method to transfer general knowledge from VFMs, gaining robust feature extraction capabilities.

This [Yao, Xufeng, et al. "Distill Vision Transformers to CNNs via Low-Rank Representation Approximation."] paper was the first to discuss knowledge distillation from ViT to CNN, proposing a representation encoding-decoding framework based on PCA and a linear transformation matrix. The framework compresses intermediate representations in the student model into low-rank latent representations through PCA, then solves for the linear transformation matrix of the decoder by maximizing the consistency between the latent representations and those of the teacher. Vision foundation models (VFMs), due to their massive parameter scale, data volume, and training paradigms, exhibit new characteristics compared to previous transformer models. This study focuses on the general representation capabilities of VFMs and uses the HGKT method to help local models at each center learn general knowledge from VFMs, achieving robust feature extraction.

To address the challenges posed by data heterogeneity in fixed layer-matching and representation shifts from input data changes, we use a meta-network to adaptively calculate model layer-matching weights and feature transfer weights, allowing for automatic and customized matching and feature transfer between VFM and CNN layers across centers. We use a learnable universal module to achieve adaptive alignment of model layer outputs. For experiments, DINOv2 serves as the teacher model, with ResNet18 and UNet as the student models, widely applied across four medical datasets (for classification and segmentation tasks). Results show VFMGL outperforms in multi-task, multi-center scenarios. Additionally, interpretability analysis confirms VFMGL's potential value for robustness and generalizability.

This [Hugo Touvron, et al. "Training data-efficient image transformers & distillation through attention" (DeiT)] paper introduces a Distillation Token, constructing a distillation loss by constraining the consistency between the Distillation Token and the teacher model's predicted labels. The Distillation Token, along with the class token, is then fed into the Transformer

encoding layers to facilitate knowledge distillation from a large CNN model to a smaller transformer model. The HGKT method proposed in this study achieves adaptive matching and general knowledge transfer between the layers of open-source VFM models and local models, providing a more flexible and detailed approach to knowledge transfer. Through this process, not only is the VFM model made more lightweight, but the local model also learns robust feature extraction by acquiring general knowledge. Furthermore, from a functional perspective, the generality of the HGKT method is not limited to Transformer-to-CNN; it is also applicable to CNN-to-Transformer, Transformer-to-Transformer, and CNN-to-CNN knowledge transfers.

Secondly, the multi-center data heterogeneity allows local models to easily fit local data while performing poorly on data from other centers. This concept, related to dataset bias, was raised by renowned scholars Antonio Torralba and Alyosha Efros in 2011 [12]. As neural networks become increasingly powerful, so does their ability to capture specific dataset patterns to improve model performance [13]. Therefore, we propose the DDBL method, which uses knowledge from the shared model to select low-heterogeneity data from each center's dataset. This selection allows the local model to learn common knowledge from the shared model, preventing the local model from learning specific data patterns and further enhancing cross-center generalization by leveraging the model's redundant parameters.

Thirdly, VFMGL demonstrates good interpretability. We conduct a feature visualization analysis on the knowledge acquired by each center's local model from the VFM in Phase I and from the shared model in Phase II. The results show that in Phase I, the learned features (Fig.10.(a)-(d)) have a stable representation capacity, with feature relationships displaying good consistency between the training and test sets. Compared to Phase I, in Phase II, the learned features (Fig.10.(e)-(h)) exhibit weaker within-center correlations, while the inter-center feature correlations become more stable (Fig.(f)(h)). This visualization supports the robustness and generalization potential of VFMGL.

Finally, the robustness and generalizability of VFMGL have been widely validated across four medical imaging tasks (classification and segmentation), including endometrial carcinoma (EC) myometrial invasion (MI) classification, breast cancer histology image classification, histological nucleus segmentation, and prostate MRI segmentation. Among them, the EC-MI dataset comprises real clinical data from multiple medical centers. VFMGL exhibits superior performance across these diverse tasks.

In summary, this study demonstrates novelty, versatility, and practicality across various aspects, including motivation, methodology, and experimentation.

- This paper introduces its distillation losses. What is the theoretical basis for these losses? How does their performance compare with state-of-the-art methods, especially the work by Hinton et al. [Hinton, Geoffrey, Oriol Vinyals, and Jeff Dean. "Distilling the knowledge in a neural network." arXiv preprint arXiv:1503.02531 (2015)]?

Response:

Thank you for the valuable comments from the reviewer. Allow me to provide a detailed introduction to our primary work here.

The heterogeneity of multi-center medical data poses challenges to model robustness and generalization, while the characteristics of vision foundation models (VFMs) show promise in mitigating this impact. This study proposes a vision foundation model general lightweight

(VFMGL) Framework that is adaptable to multiple tasks and multi-center settings. Considering the redundancy of parameters in deep learning networks, we designed this framework to allow each center to explore local model performance in stages. First, using the Heterogeneous-model General Knowledge Transfer (HGKT) method, each center extracts general knowledge from open-source VFMs to create customized local models, enabling robust feature extraction capabilities. To further improve the cross-center generalization of local models, the Data Deduction in Batch Level (DDBL) method is used to select low-heterogeneity data from each center. Through knowledge distillation (KD), the local model’s redundant parameters learn common knowledge from the shared model. The following is a detailed theoretical explanation of this framework:

First, VFMs possess large-scale model parameters trained on massive natural image datasets via self-supervised learning, endowing them with robust general representation capabilities [2]. Currently, some VFMs like DINOv2 and SAM are open-source, and even larger-scale VFMs are anticipated in the future. However, due to the distinct differences between natural and medical images, directly applying these VFMs to medical tasks is challenging [3,4], and training, fine-tuning, and deploying VFMs also encounter numerous challenges [3,4]. Therefore, we propose the HGKT method to facilitate the transfer of general knowledge from VFM to each medical center for local model development, achieving both robust feature extraction capabilities and VFM model lightweighting.

Given the disparity in knowledge domains, blindly matching model layers for knowledge transfer offers limited benefits to medical tasks and may even lead to negative transfer [14], compromising model performance. Relying on fixed model layer matching will hinder broad applicability across various medical tasks. Moreover, in multi-center medical data heterogeneity scenarios, there will be knowledge differences adapted to each center’s data [6], implying that the matching between VFM and local models will vary accordingly. Thus, we design a meta-network

ϕ_k to customize knowledge transfer for each center, which automatically calculates the model layers matching weights $\lambda_k^{m,n}$ and feature transfer weights $\omega_c^{m,n}$, providing each center with customized weights to achieve tailored model layer matching and feature transfer. Through a learnable universal module, we transform the outputs of model layers for alignment between VFM and local models. By calculating the representational differences between VFM and local models and incorporating both model layer matching weights and feature transfer weights, we construct the knowledge transfer loss (Equation 4 and Equation 5) to facilitate the local model’s learning of general knowledge. In this process, the meta-network and local model are jointly trained, with alternating updates to the local model and meta-network parameters. The weights are updated in real-time by validating the effectiveness of transferred knowledge. The model layer matching weights are used to determine the position of knowledge transfer between the VFM and the local model, while the feature transfer weights select knowledge beneficial to the task at the current center for transfer. Further details can be found in Supplementary S2-S3.

$$L_{wfm}^{m,n}(\theta_k | x, \omega^{m,n}) = \frac{1}{HW} \sum_c \omega_c^{m,n} \sum_{i,j} (r_{\theta_k}(C_{\theta_k}^n(x))_{c,i,j} - \mathbf{V}^m(x)_{c,i,j})^2 \quad (4)$$

$$L_{wfm}^{m,n}(\theta_k | x, \phi_k) = \sum_{(m,n \in \partial)} \lambda_k^{m,n} L_{wfm}^{m,n}(\theta_k | x, \omega^{m,n}) \quad (5)$$

Secondly, due to the data heterogeneity across multiple centers, local models often perform well on their own center's data but struggle with data from other centers. Studies have shown that neural networks are highly capable of capturing feature patterns specific to a particular dataset to enhance model performance [12,13]. The local model can effectively fit data with distinct, specific features on private datasets, swiftly reducing the discrepancy between the predicted and actual label distributions. However, these data samples are challenging to find similar counterparts in other centers, meaning the shared model lacks sufficient knowledge to predict such samples accurately. This leads to a growing divergence between the prediction distributions of the shared and local models. To address this, we propose the DDBL method, which selects low-heterogeneity data from each center based on shared model knowledge. With a KD approach focused on model logits, it drives the local models to learn the common knowledge shared across centers. This both suppresses the tendency of local models to learn overly specific patterns and uses redundant model parameters to further enhance cross-center generalization. More details are available in Supplementary S4-S5.

In the seminal work on knowledge distillation by Hinton, Vinyals, and Dean, Hinton et al.[7]proposed a logit-based KD method that calculates the difference in prediction probability distributions between the teacher and student models using standard cross-entropy, thereby transferring knowledge from teacher to student by minimizing this discrepancy. In the field of KD, logit-based KD and feature-based KD that uses intermediate model layers are the two primary approaches. In our research, the HGKT method serves as a feature-based KD technique, which focuses on transferring intermediate layer output features. FitNets[10] by Adriana Romero et al. ["FitNets: Hints for thin deep nets", ICLR 2015] pioneered this approach, validating the effectiveness of intermediate layer feature distillation and comparing it with logit-based KD methods. Similarly, Wang et al.[15] further substantiated the intermediate-layer KD approach in "Exclusivity-consistency regularized knowledge distillation for face recognition" (ECCV 2020). The above two studies, through fixed model layer matching, validated the performance of KD based on features from intermediate model layer outputs.

Logit-based Knowledge Distillation (KD)[7] helps the student model achieve better generalization in classification tasks by encouraging it to mimic a probability distribution similar to the teacher model., usually requiring the student and teacher models to share the same label space. To compare the performance of VFMGL with the logit-based KD method on multi-center medical data, we used Virchow and ResNet18 as the teacher and student models, respectively, for the logit-based KD method. The Virchow model [4], based on the DINOv2 [2] framework, was pre-trained on 1.5 million whole-slide histopathology images (including breast cancer pathology images) and then used model embeddings and nearly 80,000 annotated medical samples to build a classifier for specific downstream tasks. Therefore, in the breast cancer pathology image classification task (use case 2), we compared VFMGL with the logit-based KD method (Fig. S12). The results indicate that VFMGL achieved higher scores across multiple metrics on multi-center data, with stable predictive performance. Further details of the results are provided in Table S16.

Breast Cancer Metastasis Dataset(use case 2)

Fig. S12: Radar chart comparing the performance of VFMGL and the KD method based on model logic layer output.

Table S16: Performance comparison between VFMGL and the KD method based on model logic layer output.

Center	Methods	AUC	Sensitivity	Specificity	Accuracy	PPV	NPV	F1
A	VFMGL	0.9992	0.9882 (5874/5944)	0.9896 (5882/5944)	0.9889 (11756/11888)	0.9896 (5874/5936)	0.9882 (5882/5952)	0.9889
	Virchow_KD	0.9266	0.8612 (5119/5944)	0.7873 (4680/5944)	0.8243 (9799/11888)	0.8020 (5119/6383)	0.8501 (4680/5505)	0.8305
B	VFMGL	0.9973	0.9481 (3309/3490)	0.9974 (3482/3491)	0.9728 (6791/6981)	0.9973 (3309/3318)	0.9506 (3482/3663)	0.9721
	Virchow_KD	0.9981	0.9436 (3293/3490)	0.9946 (3472/3491)	0.9691 (6765/6981)	0.9943 (3293/3312)	0.9463 (3472/3669)	0.9682
C	VFMGL	0.9995	0.9911 (8430/8506)	0.9915 (8433/8505)	0.9913 (16863/17011)	0.9915 (8430/8502)	0.9911 (8433/8509)	0.9913
	Virchow_KD	0.6401	0.1743 (1483/8506)	0.9949 (8462/8505)	0.5846 (9945/17011)	0.9718 (1483/1526)	0.5465 (8462/15485)	0.2957
D	VFMGL	0.9977	0.9427 (12240/12984)	0.9988 (12969/12984)	0.9708 (25209/25968)	0.9988 (12240/12255)	0.9457 (12969/13713)	0.9699
	Virchow_KD	0.9748	0.7160 (9297/12984)	0.9970 (12945/12984)	0.8565 (22242/25968)	0.9958 (9297/9336)	0.7783 (12945/16632)	0.8331
E	VFMGL	0.9993	0.9845 (14445/14672)	0.9922 (14559/14673)	0.9884 (29004/29345)	0.9922 (14445/14559)	0.9846 (14559/14786)	0.9883
	Virchow_KD	0.9157	0.1322 (1940/14672)	0.9957 (14610/14673)	0.5640 (16550/29345)	0.9685 (1940/2003)	0.5343 (14610/27342)	0.2327

Notes: VFMGL, vision foundation model general lightweight; AUC, area under the curve; PPV, positive predictive value; NPV, negative predictive value.

- The ROC-AUC curve in Figure 2 indicates that there are not enough samples in the test set, which raises questions about the reliability of the authors' results.

Response:

We thank you for pointing out this important issue.

The VFMGL framework is suitable for multi-task, multi-center scenarios. To validate its performance, we conducted experiments on four datasets (use cases 1–4), including endometrial carcinoma (EC) myometrial invasion (MI) classification, breast cancer histology image classification, histology nuclei segmentation, and prostate MRI segmentation (data distribution is detailed in Table S8). The EC-MI classification dataset is derived from real clinical data; for this task, we collected 1,150 cases across multiple medical centers and randomly split the data into training and test sets at a 6:4 ratio. Given the varied scales of these centers, differences in data volume and distribution naturally arise, presenting a common yet essential challenge in multi-center medical scenarios.

Table S8. The distribution of data across the four datasets is illustrated.

EC-MI Classification(use case 1)						
	A	B	C	D	external center E	external center F
Train Set	410	42	22	176	-	-
Test Set	275	29	15	118	63	117
breast cancer histopathological image classification(use case 2)						
	A	B	C	D	E	
Train Set	38039	22339	54435	83096	93902	
Val Set	9509	5584	13608	20774	23475	
Test Set	11888	6981	17011	25968	29345	
prostate MRI segmentation(use case 3)						
	A	B	C	D	E	F
Train Set	167	100	299	263	245	112
Val Set	41	25	75	65	62	28
Test Set	53	32	94	83	77	35
histological cell nucleus segmentation(use case 4)						
	A	B	C	D	E	F
Train Set	47	31	44	46	24	26
Val Set	11	8	11	11	6	7
Test Set	22	22	31	26	14	17

For the EC-MI classification task, considering scenarios where individual independent centers did not participate in the VFMGL framework, we further evaluated VFMGL’s performance on external center E (external validation set 1), with data information detailed in Table 1. We calculated the data features similarity between external center E and each center within the framework (Center A: 0.15; Center B: -0.05; Center C: -0.06; Center D: 0.08) to select the optimal local model for predicting on external validation set 1. Results showed that model A achieved an AUC of 0.742 (Model B: 0.694; Model C: 0.661; Model D: 0.710).

Furthermore, based on your suggestion, we made an effort to gather an additional 117 cases from a new medical center, F, as external validation set 2 (see Table 1) to further evaluate VFMGL’s performance. Based on the above approach, we computed the similarity of data features

between centers A–D and center F, yielding values of -0.04, 0.12, -0.11, and 0.01, respectively. The results showed that model B achieved an AUC of 0.720 (model A: 0.680, model C: 0.601, model D: 0.702). The ROC curve is shown in Fig. S13. These results suggest that VFMGL maintains relatively stable predictive performance on data from new centers not involved in the information exchange.

Fig.S13 The ROC curve for VFMGL on external center F is shown.

Table 1. Patient information

Center	Set	Type	Age	FIGO stage				Histopathologic Type	
				I	II	III	IV	I	II
Center A	Train-set	NMI(49)	50.3673±9.4001	46	0	2	1	42	7
	(410)	MI(361)	55.9335±8.1279	261	31	61	8	336	25
	Test-set	NMI(34)	47.5588±7.2413	34	0	0	0	34	0
	(275)	MI(241)	54.8423±7.8782	168	21	50	2	216	25
Center B	Train-set	NMI(8)	48.3750±5.8294	8	0	0	0	8	0
	(42)	MI(34)	52.8824±7.9839	28	2	3	1	32	2
	Test-set	NMI(6)	45.5000±10.4451	6	0	0	0	6	0
	(29)	MI(23)	55.5652±10.2772	16	3	4	0	22	1
Center C	Train-set	NMI(1)	—	1	0	0	0	0	1
	(22)	MI(21)	60.9524±8.1944	15	2	3	1	19	2
	Test-set	NMI(1)	—	1	0	0	0	1	0
	(15)	MI(14)	60.1429±10.2421	9	3	2	0	14	0

Center D	Train-set	NMI(11)	51.1818±11.7883	11	0	0	0	10	1
	(176)	MI(165)	54.6788±8.1195	127	5	31	2	148	17
	Test-set	NMI(8)	51.0000±3.1168	7	0	1	0	8	0
	(118)	MI(110)	54.4636±7.8503	91	3	14	2	101	9
External validation center E	Test-set	NMI(1)	—	1	0	0	0	1	0
	(63)	MI(62)	55.0968±8.9511	42	11	7	2	60	2
External validation center F	Test-set	NMI(8)	54.6250±13.7730	8	0	0	0	7	1
	(117)	MI(109)	54.9358±8.9134	91	2	15	1	99	10

- The motivation for this study might not be valid. The size of modern transformers is not as cumbersome as suggested by the authors. For example, Swin-T models have roughly the same size of parameters as ResNet-50.

Response:

Thank you for your suggestions. We realize that our initial wording may have led to some misunderstanding of our research motivation, so please allow me to clarify our work and objectives here.

Compared to previous transformer models (such as Swin-T), vision foundation models (VFMs) based on transformers benefit from extensive parameter scales, data volume, and self-supervised training paradigms, thereby demonstrating powerful general representation capabilities. Our study focuses on these general representation abilities of VFMs. The HGKT method facilitates robust feature extraction by automatically matching model layers and adaptively transferring general knowledge from open-source VFMs to local models. Additionally, as previously mentioned, the versatility of HGKT is not restricted to Transformer-to-CNN transfer; it is also applicable for transferring knowledge from VFMs to lightweight transformer structures. Therefore, VFMGL exhibits practical applicability and generalizability.

References

- [1] Kirillov A , Mintun E , Ravi N ,et al. Segment Anything. 2023 IEEE/CVF International Conference on Computer Vision (ICCV). 3992-4003 (2023).
- [2] Maxime Oquab, Timothée Darcet, Théo Moutakanni. et al. DINOv2: Learning Robust Visual Features without Supervision. Preprint at <https://arxiv.org/pdf/2304.07193> (2023)
- [3] Ma, J., He, Y., Li, F. et al. Segment anything in medical images. Nat Commun 15, 654 (2024). <https://doi.org/10.1038/s41467-024-44824-z>
- [4] Vorontsov, E., Bozkurt, A., Casson, A. et al. A foundation model for clinical-grade computational pathology and rare cancers detection. Nat Med 30, 2924–2935 (2024). <https://doi.org/10.1038/s41591-024-03141-0>
- [5] Chen YQ, Lu W, Qin X, Wang JD and Xie X. MetaFed: Federated Learning among Federations with Cyclic Knowledge Distillation for Personalized Healthcare. IJCAI'22 federated learning workshop.

- [6] Jiang MR, Wang ZR, Dou Q. HarmoFL: Harmonizing Local and Global Drifts in Federated Learning on Heterogeneous Medical Images. Proceedings of the 36th AAAI Conference on Artificial Intelligence. 36, 914-922 (2022).
- [7] Hinton, Geoffrey, O. Vinyals, and J. Dean. Distilling the Knowledge in a Neural Network. NIPS 2014.
- [8] Han, Song, H. Mao, and W. J. Dally. Deep Compression: Compressing Deep Neural Networks with Pruning, Trained Quantization and Huffman Coding. ICLR 2016.
- [9] K. Zhu, X. Hu, J. Wang, et al. Improving Generalization of Adversarial Training via Robust Critical Fine-Tuning. 2023 IEEE/CVF International Conference on Computer Vision (ICCV). 4401-4411 (2023).
- [10] Romero, A., Ballas, N., Kahou, S. E., Chassang, A., Gatta, C., & Bengio, Y. Fitnets: Hints for thin deep nets. ICLR 2015.
- [11] Wang X, Fu T, Liao S, et al. Exclusivity-Consistency Regularized Knowledge Distillation for Face Recognition[C]//European Conference on Computer Vision. Springer, Cham, 2020.DOI:10.1007/978-3-030-58586-0_20.
- [12] Antonio Torralba and Alexei A Efros. Unbiased look at dataset bias. CVPR 2011.
- [13] Zhuang Liu, Kaiming He. A Decade's Battle on Dataset Bias: Are We There Yet? <https://doi.org/10.48550/arXiv.2403.08632>
- [14] Pan S J, Yang Q. A survey on transfer learning[J]. IEEE Transactions on knowledge and data engineering, 2010, 22(10): 1345-1359.
- [15] Wang X, Fu T, Liao S, et al. Exclusivity-Consistency Regularized Knowledge Distillation for Face Recognition[C]//European Conference on Computer Vision. Springer, Cham, 2020.DOI:10.1007/978-3-030-58586-0_20.

REVIEWERS' COMMENTS

Reviewer #3 (Remarks to the Author):

I think the authors had carefully replied to all my comments previously. I have nothing to add.

Response:

Thank you very much for your positive feedback. We are grateful for your review, and we appreciate the opportunity to improve our manuscript.

Reviewer #4 (Remarks to the Author):

The authors have provided a comprehensive response to the reviewers' comments, addressing all points effectively.

Response:

Thank you very much for your positive feedback and thoughtful review.

Reviewer #5 (Remarks to the Author):

What are the noteworthy results?

The authors propose a light-weight framework adapted to Vision Foundation Models in a medical image analysis setup.

Response:

Thank you very much for your positive feedback and thoughtful review.

Will the work be of significance to the field and related fields? How does it compare to the established literature? If the work is not original, please provide relevant references.

No. The proposed method makes incremental contributions by addressing engineering challenges when using Vision Foundation Models in multi-center setups. The scientific problem behind is not that significant. Additionally, the proposed method shall also work in non-medical scenarios.

Response:

Thank you for your thoughtful review and recognition of our research. The heterogeneity among different medical centers in terms of imaging equipment, imaging protocols, image quality, and patient demographics presents challenges in the medical image data heterogeneity across centers. Data heterogeneity affects the stability of feature extraction, as well as the model's generalization ability and robustness. In response to the impact of multi-center data heterogeneity on model performance, this study proposes the Vision Foundation Model General Lightweight (VFMGL) framework. VFMGL transfers general knowledge from the vision foundation model,

achieving model lightweighting while improving the robustness of the lightweight model. It then learns common knowledge from the shared model to further enhance the cross-center generalization performance of the lightweight model.

Does the work support the conclusions and claims, or is additional evidence needed?

Yes.

Response:

Thank you very much for your positive feedback and thoughtful review.

Are there any flaws in the data analysis, interpretation and conclusions?

Do these prohibit publication or require revision?

Not found. The authors have extensively revised the manuscript in terms of additional results.

Response:

Thank you for your thoughtful review and recognition of this research.

Is the methodology sound? Does the work meet the expected standards in

your field?

The work does show merit for a computer science or engineering journal. But for Nature Communications, the scientific and practical clinical value requires further justifications. Based on my understanding, the proposed method is evaluated based on simulation. No real system is deployed yet.

Response:

Thank you for your suggestions and thoughtful review. In this study, we validated the generalizability and practicality of the Vision Foundation Model General Lightweight (VFMGL) framework across four medical tasks (classification and segmentation). VFMGL is capable of constructing lightweight and robust models by adaptively transferring general knowledge in multi-medical-center settings, achieving the lightweighting of vision foundation models. We conducted experiments in the contexts of endometrial cancer (EC) myometrial invasion (MI) classification task, breast cancer histology image classification task, histological nucleus segmentation task, and prostate MRI segmentation task, and verified the robustness and generalization of VFMGL.

Is there enough detail provided in the methods for the work to be reproduced?

I believe so, but I didn't check the details.

Response:

Thank you very much for your positive feedback and thoughtful review. To enhance the reproducibility of this work, we have made the relevant code available on GitHub (<https://github.com/baofengguat/VFMGL/tree/main>) and provided the corresponding data source files for the figures and tables in the manuscript and supplementary materials (<https://pan.baidu.com/s/1hLyEr43F8jJ8uj58i-Xueg?pwd=zmvy>, access code: zmvy).